# SSIF: Physics-Inspired Implicit Representations for Spatial-Spectral Image Super-Resolution

## Abstract

Existing digital sensors capture images at fixed spatial and spectral resolutions (e.g., RGB, multispectral, and hyperspectral images), and generating super-resolution images with different resolution settings requires bespoke machine learning models. Spatial Implicit Functions (SIFs) partially overcome the spatial resolution challenge by representing an image in a spatial-resolution-independent way. However, they still operate at fixed, pre-defined spectral resolutions. To address this challenge, we propose **Spatial-Spectral Implicit Function (SSIF)**, a neural implicit model that represents an image as a function of both continuous pixel coordinates in the spatial domain and continuous wavelengths in the spectral domain. This continuous representation across spatial and spectral domains enables *a single model to learn from a diverse set of resolution settings*, which leads to better generalizability. This representation also allows the *physical principle of spectral imaging* and the spectral response functions of sensors to be easily incorporated during training and inference. Moreover, SSIF does not have the equal spectral wavelength interval requirement for both input and output images which leads to much better applicability. We empirically demonstrate the effectiveness of SSIF on two challenging spatial-spectral super-resolution benchmarks. We observe that SSIF consistently outperforms state-of-the-art baselines even when the baselines are allowed to train separate models at each spatial or spectral resolution. We show that SSIF generalizes well to both unseen spatial and spectral resolutions. Moreover, due to its physics-inspired design, SSIF performs significantly better at low data regime and converges faster during training compared with other strong neural implicit function-based baselines.

## 1 Introduction

While the physical world is continuous, most digital sensors (e.g., cell phone cameras, multispectral or hyperspectral sensors in satellites) can only capture a discrete representation of continuous signals in both spatial and spectral domains (i.e., with a fixed number of spectral bands, such as red, green, and blue). Due to the limited energy of incident photons, fundamental limitations in achievable signal-to-noise ratios (SNR), and time constraints, there is always a trade-off between spatial and spectral resolution (Mei et al., 2020; Ma et al., 2021)[1]. High spatial resolution and high spectral resolution can not be achieved at the same time, leading to a variety of spatial and spectral resolutions used in practice for different sensors. However, ML models are typically bespoke to certain resolutions, and models typically do not generalize to spatial or spectral resolutions they have not been trained on. This calls for image super-resolution (SR) methods, which are capable of increasing the spatial or spectral resolution of a given single low-resolution image (Galliani et al., 2017). It has become increasingly important for a wide range of tasks including object recognition and tracking (Pan et al., 2003; Uzair et al., 2015; Xiong et al., 2020), medical image processing (Lu & Fei, 2014; Johnson et al., 2007), remote sensing (He et al., 2021b; Bioucas-Dias et al., 2013; Melgani & Bruzzone, 2004; Zhong et al., 2018; Wang et al., 2022a; Liu et al., 2023), and astronomy (Ball et al., 2019).

The diversity in input-output image resolutions (both spatial and spectral) significantly increases the complexity of deep neural network (DNN) based SR model development. Most SR research develops

---

[1]Given a fixed overall sensor size and exposure time, higher spatial resolution and higher spectral resolution require the per pixel sensor to be smaller and bigger at the same time, which are contradicting each other.

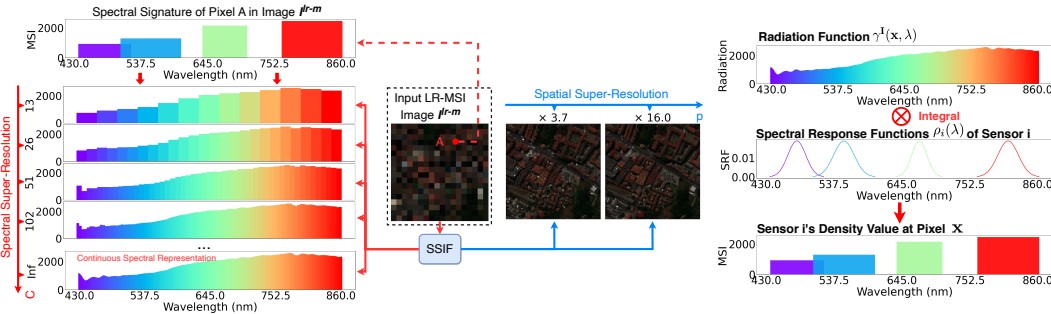

(a) Spatial-Spectral Implicit Function (SSIF)  (b) The physical principle of spectral imaging

Figure 1: (a) SSIF represents an input low-resolution multispectral (LR-MSI) image $\mathbf{I}^{lr-m}$ as a continuous function $\gamma_{\mathbf{I}}(\mathbf{x}, \lambda)$ on both pixel coordinates $\mathbf{x}$ in the spatial domain and wavelengths $\lambda$ in the spectral domain. SSIF can perform both spatial (blue arrows) and spectral (red arrows) super-resolution simultaneously (illustrated with a specific pixel A). (b) An illustration of the physical principle of spectral imaging for MSI and HSI sensors.

separate DNN models for each input-output image resolution pairs with a specific spatial and spectral resolution (Lim et al., 2017; Zhang et al., 2018b; Ma et al., 2021; Mei et al., 2020; Ma et al., 2022). For example, convolution-based SR models such as RCAN (Zhang et al., 2018a), SR3(Saharia et al., 2021), SSJSR (Mei et al., 2020), (He et al., 2021b), and SSFIN (Ma et al., 2022) need to be trained separately for each input-output image resolution settings[2]. This practice has three limitations: *1) For some SR settings with much less training data, these models can yield suboptimal results or lead to overfitting; 2) It prevents generalizing trained SR models to unseen spatial/spectral resolutions. 3) it is hard to incorporate domain knowledge such as sensor response functions into the model design.* Inspired by the recent progress in 3D reconstruction with implicit neural representation (Park et al., 2019; Mescheder et al., 2019; Chen & Zhang, 2019; Sitzmann et al., 2020; Mildenhall et al., 2020), image neural implicit functions (NIF) (Dupont et al., 2021; Chen et al., 2021; Yang et al., 2021; Zhang, 2021; Cao et al., 2023) partially overcome the aforementioned problems (especially the second one) by learning a continuous function that maps an arbitrary pixel spatial coordinate to the corresponding visual signal value and generate images at any spatial resolution. We call them *Spatial Implicit Functions (SIF)*. However, each SIF model still has to be trained separately to target a specific spectral resolution (i.e., a fixed number of spectral bands).

Extending SIFs to the spectral domain is a non-trivial task due to the complexities of the spectral response functions. First, the response functions of different bands might not be simple functions (e.g., Gaussian or more complicated functions) and can be different types. Second, the bands of the input/output images might be unequally spaced in the spectral domain. For many RGB or multispectral images, each band may have different spectral widths (i.e., lengths of wavelength intervals) and different bands' wavelength intervals may even overlap with each other. The "Spectral Signature of Pixel A" of the image $\mathbf{I}^{lr-m}$ in Figure 1a shows one example of such cases. Recent work like LISSF (Zhang et al., 2024) utilizes 3D CNN in the image encoder to naively generalize SIFs into a spatial-spectral SR model. However, LISSF relies on a strong assumption that all input images should have equal-spaced spectral wavelength intervals which most RGB and multispectral images do not satisfy. This significantly limits its applicability in most spatial-spectral SR problems. Therefore, effectively incorporating images from various sensors with diverse characteristics is the key to achieving cost-effectiveness and model generalizability, but poses a great challenge to modeling.

In this work, we propose Spatial-Spectral Implicit Function ($SSIF$), which generalizes neural implicit representations to the spectral domain as a physics-inspired architecture by incorporating sensors' physical principles of spectral imaging (Nguyen et al., 2014; Zheng et al., 2020). $SSIF$ represents an image $\mathbf{I}$ as a continuous function $\gamma^{\mathbf{I}}(\mathbf{x}, \lambda)$ on both pixel spatial coordinates $\mathbf{x}$ in the spatial domain and wavelengths $\lambda$ in the spectral domain. As shown in Figure 1a, given an input low-resolution multispectral (or RGB) image, a single $SSIF$ model can generate images with different spatial and spectral resolutions. To tackle the problem of modeling response functions $\rho_i(\lambda)$ of sensor $i$, we predict each spectral band value of each target pixel $\mathbf{x}$ as the integral of the radiation function $\gamma^{\mathbf{I}}(\mathbf{x}, \lambda)$ of pixel $\mathbf{x}$ and the response function $\rho_i(\lambda)$ (see Figure 1b as an illustration). Our contributions are as follows:

---

[2]Figure 9a in Appendix A.1 illustrates this separate training practice.

1. We propose $SSIF$ which represents an image as a physics-inspired continuous function on both pixel coordinates in the spatial domain and wavelengths in the spectral domain. Unlike LISSF, $SSIF$ does not have the equally spaced spectral band requirement for both input and output images. It can handle SR tasks with different spatial and spectral resolutions simultaneously.
2. We demonstrate the effectiveness of $SSIF$ on two challenging spatial-spectral super-resolution benchmarks – CAVE (the indoor scenes) and Pavia Centre (Hyperspectral Remote Sensing images). SSIF consistently outperforms state-of-the-art SR baseline models on spatial SR, spectral SR, and spatial-spectral SR tasks even when the baselines are trained separately at each spectral resolution (and spatial resolution). We show that SSIF generalizes well to both unseen spatial and spectral resolutions.
3. We show that due to the physics-inspired design – explicitly incorporating physical principles of spectral imaging into SSIF's model design, SSIF performs significantly better at low data regime and converges faster during training compared with existing SIF baselines.

## 2 RELATED WORK

**Image Super Resolution**   As an ill-posed single image-to-image translation problem, super-resolution (SR) aims at increasing the spatial or spectral resolution of a given image such that it can be used for different downstream tasks. It has been widely used on natural images(Zhang et al., 2018a; Hu et al., 2019; Zhang et al., 2020b; Saharia et al., 2021; Chen et al., 2021), screen-shot images (Yang et al., 2021), omnidirectional images (Deng et al., 2021; Yoon et al., 2021) medical images (Isaac & Kulkarni, 2015), as well as multispectral (He et al., 2021b; Liu et al., 2023) and hyperspectral remote sensing images(Mei et al., 2017; Ma et al., 2021; Mei et al., 2020; Wang et al., 2022b). Traditionally image SR (Ledig et al., 2017; Lim et al., 2017; Zhang et al., 2018b; Haris et al., 2018; Zhang et al., 2020c; Yao et al., 2020; Mei et al., 2020; Saharia et al., 2021; Ma et al., 2021; He et al., 2021b; Ma et al., 2022; Cao et al., 2023) has been classified into three tasks according to the input and output image resolutions:[3] Spatial Super-Resolution (spatial SR), Spectral Super-Resolution (spectral SR) and Spatio-Spectral Super-Resolution (SSSR). Spatial SR (Zhang et al., 2018a; Hu et al., 2019; Zhang et al., 2020a; Niu et al., 2020; Wu et al., 2021b; Chen et al., 2021; He et al., 2021b) focuses on increasing the spatial resolution of the input images (e.g., from $h \times w$ pixels to $H \times W$ pixels) while keeping the spectral resolution (*i.e.*, number of spectral bands/channels) unchanged. In contrast, spectral SR (Galliani et al., 2017; Fu et al., 2018; Arad et al., 2018; Kaya et al., 2019; Fu et al., 2020; He et al., 2021a; Sun et al., 2021; Zhu et al., 2021; Zhang, 2021; Mei et al., 2022; Zhang et al., 2022; He et al., 2023) focuses on increasing the spectral resolution of the input images (e.g., from $c$ to $C$ channels) while keeping the spatial resolution fixed[4]. SSSR (Mei et al., 2020; Ma et al., 2021; 2022) focuses on increasing both the spatial and spectral resolution of the input images. Here, $h, w$ (or $H, W$) indicates the height and width of the low-resolution, LR, (or high-resolution, HR) images while $c$ and $C$ indicate the number of bands/channels of the low/high spectral resolution images. For video signal, SR can also be done along the time dimension, but we don't consider it here and leave it as future work.

**Implicit Neural Representation**   Recently, we have witnessed an increasing amount of work using implicit neural representations for different tasks such as image regression (Tancik et al., 2020) and compression(Dupont et al., 2021; Strümpler et al., 2021), 3D shape regression/reconstruction (Mescheder et al., 2019; Tancik et al., 2020; Chen & Zhang, 2019), 3D shape reconstruction via image synthesis (Mildenhall et al., 2020), 3D magnetic resonance imaging (MRI) reconstruction (Tancik et al., 2020), 3D protein reconstruction (Zhong et al., 2020), spatial feature distribution modeling (Mai et al., 2020b; 2022; 2023b; Cole et al., 2023; Mai et al., 2023a; Rußwurm et al., 2024; Wu et al., 2024), geographic question answering (Mai et al., 2020a), and etc. The core idea is to learn a continuous function that maps spatial coordinates (e.g., pixel coordinates, 3D coordinates, and geographic coordinates) to the corresponding signals (e.g., point cloud intensity, MRI intensity, visual signals, etc.). A common setup is to input the spatial coordinates in a deterministic or learnable Fourier feature mapping layer (Tancik et al., 2020) (consisting of sinusoidal functions with different frequencies), which converts the coordinates into multi-scale features. Then a multi-layer perceptron

---

[3]A related task, Multispectral and Hyperspectral Image Fusion (Zhang et al., 2020c; Yao et al., 2020), takes a high spatial resolution MSI image and a low spatial resolution HSI image as inputs and generates a high-resolution HSI image. Here, we focus on the single image-to-image problem and leave this as future work.

[4]See He et al. (2023); Zhang et al. (2022) for comprehensive reviews on different deep-learning-based spectral SR models.

further transforms these multi-scale features for downstream tasks. In parallel, **neural implicit functions (NIF)** such as LIIF (Chen et al., 2021), ITSRN (Yang et al., 2021), Zhang (2021), and CiaoSR (Cao et al., 2023) are proposed for image spatial SR which map pixel spatial coordinates to the visual signals in the high spatial resolution images. One outstanding advantage is that they can jointly handle spatial SR tasks at an arbitrary spatial scale. Recently, LISSF (Zhang et al., 2023; 2024) was developed as a NIF-based SSSR model that uses an image encoder with 3D CNN and generalizes LIIF with 3D coordinates in spatial and spectral space for arbitrary scale SSSR. However, it adopts a strong assumption that input images' bands must have equally spaced spectral wavelength intervals which most RGB and multispectral images do not satisfy. This significantly limits LISSF's applicability. In all, to our best knowledge, the existing NIF-based models learn continuous image representations in the spatial domain while still operating either at fixed pre-defined spectral resolutions, or on input images with equally spaced wavelength intervals. In comparison, our SSIF can make predicsions for sensors with arbitrary response functions by leveraging physical characteristics for the light sources and sensors. Both input and output images of SSIF can have irregularly spaced wavelength intervals with arbitrary upsampling spectral scales.

## 3 PROBLEM STATEMENT

The spatial-spectral image super-resolution (SSSR) problem over various spatial and spectral resolutions can be conceptualized as follows. Given an input low spatial/spectral resolution (LR-MSI) image $\mathbf{I}^{lr-m} \in \mathbb{R}^{h \times w \times c}$, we want to generate a high spatial and spectral resolution (HR-HSI) image $\mathbf{I}^{hr-h} \in \mathbb{R}^{H \times W \times C}$. Here, $h, w, c$ and $H, W, C$ are the height, width and channel dimension of image $\mathbf{I}^{lr-m}$ and $\mathbf{I}^{hr-h}$, and $H > h$, $W > w$, $C > c$. The spatial upsampling scale $p$ is defined as $p = H/h = W/w$. Without loss of generality, let $\Lambda^{hr-h} = [\Lambda_0^T, \Lambda_1^T, ..., \Lambda_C^T] \in \mathbb{R}^{C \times 2}$ be the wavelength interval matrix, which defines the spectral bands in the target HR-HSI image $\mathbf{I}^{hr-h}$. Here, $\Lambda_i = [\lambda_{i,s}, \lambda_{i,e}] \in \mathbb{R}^2$ is the wavelength interval for the $i$th band of $\mathbf{I}^{hr-h}$ where $\lambda_{i,s}, \lambda_{i,e}$ are the start and end wavelength of this band. $\Lambda^{hr-h}$ can be used to fully express the spectral resolution of the target HR-HSI image $\mathbf{I}^{hr-h}$. In this work, we do not use $C/c$ to represent the spectral upsampling scale because bands/channels of image $\mathbf{I}^{lr-m}$ and $\mathbf{I}^{hr-h}$ might not be equally spaced (See Figure 1a). So $\Lambda^{hr-h}$ is a very flexible representation for the spectral resolution, capable of representing situations when different bands have different spectral widths or their wavelength intervals overlap with each other. When $\mathbf{I}^{hr-h}$ has equally spaced wavelength intervals, such as those of most of the hyperspectral images, we use its band number $C$ to represent the spectral scale.

The spatial-spectral super-resolution (SSSR) can be represented as a function

$$\mathbf{I}^{hr-h} = H^{sr}(\mathbf{I}^{lr-m}, p, \Lambda^{hr-h}) \tag{1}$$

where $H^{sr}(\cdot)$ takes as input the image $\mathbf{I}^{lr-m}$, the desired spatial upsampling scale $p$, and the target sensor wavelength interval matrix $\Lambda^{hr-h}$, and generates the HR-HSI image $\mathbf{I}^{hr-h} \in \mathbb{R}^{H \times W \times C}$. In other words, we aim at learning **one single function** $H^{sr}(\cdot)$ that can take any input images $\mathbf{I}^{lr-m}$ with a fixed spatial and spectral resolution, and generate images $\mathbf{I}^{hr-h}$ with diverse spatial and spectral resolutions specified by different $p$ and $\Lambda^{hr-h}$.

Note that none of the existing SR models can achieve this. Most classic SR models have to learn separate $H^{sr}(\cdot)$ for different pairs of $p$ and $\Lambda^{hr-h}$ such as EDSR Lim et al. (2017), RCAN (Zhang et al., 2018a), SR3(Saharia et al., 2021), SSJSR (Mei et al., 2020), He et al. (2021b), SwinIR (Liang et al., 2021), and SSFIN (Ma et al., 2022). For SIF models such as LIIF(Chen et al., 2021), SADN (Wu et al., 2021a), ITSRN (Yang et al., 2021), Zhang (2021), CiaoSR (Cao et al., 2023), they can learn one $H^{sr}(\cdot)$ for different $p$ but with a fixed $\Lambda^{hr-h}$ (see Figure 9). LISSF (Zhang et al., 2024) can learn one $H^{sr}(\cdot)$ for different $p$ and $\Lambda^{hr-h}$ but it requires the wavelength interval matrix $\Lambda^{lr-m} \in \mathbb{R}^{c \times 2}$ of $\mathbf{I}^{lr-m}$ equally spaced while SSIF allows arbitrary $\Lambda^{lr-m}$.

## 4 SPATIAL-SPECTRAL IMPLICIT FUNCTION

In order to achieve generalizability we design SSIF based on light sensor and light source principles.

### 4.1 LIGHT SENSOR PRINCIPLE

On the sensor side, the SSIF model design follows the physical principle that the pixel density value of a sensor can be computed by an integral of the radiance function $\gamma^I(x, \lambda)$ and the response function $\rho(\lambda)$ of a sensor. More specifically, let $\mathbf{s}_{\mathbf{x},i}$ be the pixel density value of a pixel $\mathbf{x}$ at the spectral band

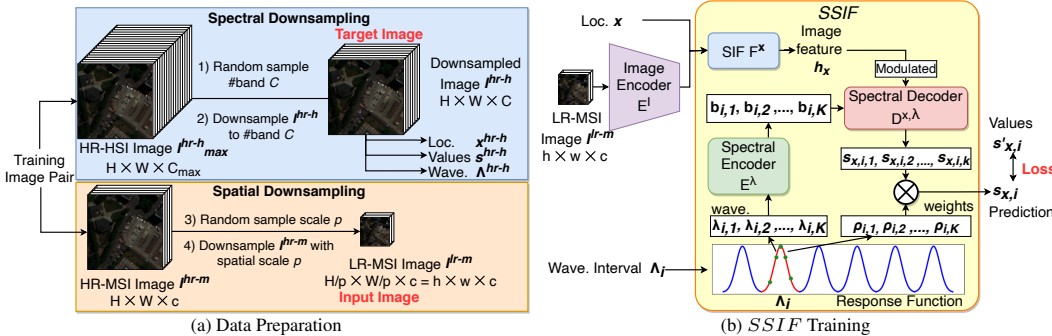

(a) Data Preparation  (b) $SSIF$ Training

Figure 2: Data preparation (a) and training (b) for $SSIF$. In Figure (b), we use Gaussian distributions as the response functions for different wavelength intervals $\{\Lambda_1, \Lambda_2, .., \Lambda_C\}$ while the response function $\rho_i(\lambda_{i,k})$ for $\Lambda_i$ is highlighted in red. The green dots are $K$ wavelengths $\{\lambda_{i,1}, \lambda_{i,2}, ..., \lambda_{i,K}\}$ sampled from a wavelength interval $\Lambda_i = [\lambda_{i,s}, \lambda_{i,K}] \in \Lambda^{hr-h}$ and $\{\rho_{i,1}, \rho_{i,2}, ..., \rho_{i,K}\}$ are their corresponding response function values. $\{\mathbf{b}_{i,1}, \mathbf{b}_{i,2}, ..., \mathbf{b}_{i,K}\}$ are their encoded spectral embeddings. $\otimes$ represents the weighted sum in Equation 6.

$b_i$ with wavelength interval $\Lambda_i$. It can be computed by an integral of the **radiance function** $\gamma^{\mathbf{I}}(\mathbf{x}, \lambda)$ and **response function** $\rho_i(\lambda)$ of a sensor at band $b_i$ (see Figure 1b as an illustration).

$$\mathbf{s}_{\mathbf{x},i} = \int_{\Lambda_i} \rho_i(\lambda)\gamma^{\mathbf{I}}(\mathbf{x}, \lambda) \, \mathrm{d}\lambda \tag{2}$$

where $\lambda$ is wavelength. So for each pixel $\mathbf{x}$, the radiance function is a neural field that describes the radiance curve as a function of the wavelength. Note that, unlike recent NeRF where only three discrete wavelength intervals (i.e., RGB) are considered, we aim to learn a *continuous* radiance curve over wavelength for each pixel. The spectral response function (Zheng et al., 2020) describes the sensitivity of the sensor to different wavelengths and is usually sensor-specific. For example, the red sensor in commercial RGB cameras has a strong response (i.e., high pixel density) to red light. The spectral response functions of many commercial hyperspectral sensors (e.g., AVIRIS's ROSIS-03[5], EO-1 Hyperion) are very complex due to atmospheric absorption. A common practice adopted by many studies (Barry et al., 2002; Brazile et al., 2008; Cundill et al., 2015; Crawford et al., 2019; Chi et al., 2021) is to approximate the response functions of individual spectral bands as a Gaussian distribution or a uniform distribution. In this work, we adopt this practice and show that this inductive bias enforced via physical laws improves generalization.

## 4.2 LIGHT SOURCE PRINCIPLE

On the light source side, SSIF model design leverages the "spectral signature" principle that *the spectral intensity curve (radiation as a function of wavelength) of any pixel $\gamma^{\mathbf{I}}(\mathbf{x}, \lambda)$ can be decomposed as a weighted sum of k pretrained spectral signature functions.* This constraint enforces a useful regularity that different surface types such as water, bare ground, and vegetation reflect radiation differently in various wavelengths and have their unique spectral signatures[6]. With the decomposition of the pixel spectral intensity curve as a weighted sum of learnable spectral signature functions, it is possible to learn them from raw data, which often contains mixed surface types.

## 4.3 SSIF ARCHITECTURE

In the following, we will discuss the design of our SSIF which allows us to train a single SR model for different $p$ and $\Lambda^{hr-h}$. The whole model architecture of SSIF is illustrated in Figure 2b.

Following previous SIF works (Chen et al., 2021; Yang et al., 2021; Cao et al., 2023), SSIF first uses an image encoder $E^I(\cdot)$ to convert the input image $\mathbf{I}^{lr-m} \in \mathbb{R}^{h \times w \times c}$ into a 2D feature map $\mathbf{S}^{lr-m} = E^I(\mathbf{I}^{lr-m}) \in \mathbb{R}^{h \times w \times d^I}$ which shares the same spatial shape as $\mathbf{I}^{lr-m}$ but with a larger channel dimension. $E^I(\cdot)$ can be any convolution-based image encoder such as EDSR (Lim et al., 2017), RDN (Zhang et al., 2018b), or SwinIR (Liang et al., 2021). Then we can approximate the integral of Equation 2 as a weighted sum over the predicted radiance values of $K$ wavelengths $\{\lambda_{i,1}, \lambda_{i,2}, ..., \lambda_{i,K}\}$ sampled from a wavelength interval $\Lambda_i = [\lambda_{i,s}, \lambda_{i,e}] \in \Lambda^{hr-h}$ at location $\mathbf{x}$

---

[5]https://crs.hi.is/?page_id=877
[6]https://www.esa.int/SPECIALS/Eduspace_EN/SEMPNQ3Z2OF_2.html

$$\mathbf{s}_{\mathbf{x},i} = \sum_{k=1}^{K} \rho_i(\lambda_{i,k}) \gamma^{\mathbf{I}}(\mathbf{x}, \lambda_{i,k}) = \sum_{k=1}^{K} \rho_i(\lambda_{i,k}) G^{\mathbf{x},\lambda}(\mathbf{S}^{lr-m}, \mathbf{x}, \lambda_{i,k}) \tag{3}$$

Here, $\rho_i(\lambda)$ is the response function value, i.e., weight, of wavelength $\lambda$ given the current response function for band $b_i$. $\gamma^{\mathbf{I}}(\mathbf{x}, \lambda)$ is the radiance value of $\lambda$ at location $\mathbf{x}$ which can be computed by a neural implicit function $G^{\mathbf{x},\lambda}$, which maps an arbitrary pixel location $\mathbf{x} \in [-1, 1] \odot [-1, 1]$ of $\mathbf{I}^{hr-h}$ and a wavelength $\lambda_{i,k} \in \Lambda_i$ into the radiance value of the target image $\mathbf{I}^{hr-h}$ at the corresponding location and wavelength, i.e., $\gamma^{\mathbf{I}}(\mathbf{x}, \lambda_{i,k}) = G^{\mathbf{x},\lambda}(\mathbf{S}^{lr-m}, \mathbf{x}, \lambda_{i,k})$. Here, $\odot$ is the Cartesian product.

$G^{\mathbf{x},\lambda}$ can be decomposed into three neural implicit functions – a pixel feature decoder $F^{\mathbf{x}}$, a spectral encoder $E^{\lambda}$, and a spectral decoder $D^{\mathbf{x},\lambda}$. The pixel feature decoder takes the 2D feature map of the input image $\mathbf{S}^{lr-m}$ as well as one arbitrary pixel location $\mathbf{x} \in [-1, 1] \odot [-1, 1]$ of $\mathbf{I}^{hr-h}$ and maps them to a pixel hidden feature $\mathbf{h}_{\mathbf{x}} \in \mathbb{R}^d$ where $d$ is the hidden pixel feature dimension (see Equation 4). Here, $F^{\mathbf{x}}$ can be any spatial implicit function such as LIIF Chen et al. (2021), ITSRN (Yang et al., 2021), and CiaoSR (Cao et al., 2023).

$$\mathbf{h}_{\mathbf{x}} = F^{\mathbf{x}}(\mathbf{S}^{lr-m}, \mathbf{x}) \tag{4}$$

The spectral encoder $E^{\lambda}$ encodes a wavelength $\lambda_{i,k}$ sampled from any wavelength interval $\Lambda_i = [\lambda_{i,s}, \lambda_{i,e}] \in \Lambda^{hr-h}$ into a spectral embedding $\mathbf{b}_{i,k} \in \mathbb{R}^d$. We can implement $E^{\lambda}$ as any position encoder (Vaswani et al., 2017; Mai et al., 2020b). Please refer to Appendix A.2 for a detailed description. Here, we will sample $K$ wavelength from each $\Lambda_i$ according to its spectral response function as shown in Figure 2b.

$$\mathbf{b}_{i,k} = E^{\lambda}(\lambda_{i,k}) \tag{5}$$

Finally, the spectral decoder $D^{\mathbf{x},\lambda}$ maps the image feature embedding $\mathbf{h}_{\mathbf{x}}$ and the spectral embedding $\mathbf{b}_{i,k}$ into a radiance value $\mathbf{s}_{\mathbf{x},i,k} = D^{\mathbf{x},\lambda}(\mathbf{b}_{i,k}; \mathbf{h}_{\mathbf{x}})$ for $\lambda_{i,k}$ at location $\mathbf{x}$. So we have the prediction

$$\mathbf{s}_{\mathbf{x},i} = \sum_{k=1}^{K} \rho_i(\lambda_{i,k}) \mathbf{s}_{\mathbf{x},i,k} = \sum_{k=1}^{K} \rho_i(\lambda_{i,k}) D^{\mathbf{x},\lambda}(\mathbf{b}_{i,k}; \mathbf{h}_{\mathbf{x}}) \tag{6}$$

$D^{\mathbf{x},\lambda}$ can be implemented as different NN architectures. Our ablation study (see Figure 14 in Appendix A.9.2) shows that **a simple dot product function, which satisfies the "spectral signature" principle, performs very well**. The response function $\rho_i(\lambda_{i,k})$ can be a learnable function or a predefined function depending on the target HSI sensor. For this study, we use predefined functions, e.g. a Gaussian distribution or a uniform distribution, for each band $b_i$ by following Chi et al. (2021).

For training, the prediction $\mathbf{s}_{\mathbf{x},i} \in \mathbb{R}^C$ is compared with the ground truth $\mathbf{s}'_{\mathbf{x},i}$ using a L1 loss:

$$\mathcal{L} = \sum_{(\mathbf{I}^{lr-m}, \mathbf{I}^{hr-h}) \in \mathcal{D}} \sum_{(\mathbf{x}, \mathbf{s}^{hr-h}, \Lambda^{hr-h}) \in \mathbf{I}^{hr-h}} \sum_{\Lambda_i \in \Lambda^{hr-h}} \| \mathbf{s}_{\mathbf{x},i} - \mathbf{s}'_{\mathbf{x},i} \|_1 \tag{7}$$

Here the dataset $\mathcal{D}$ contains all the low-res and high-res image pairs for the SSSR task. Figure 2a illustrates the data preparation process of SSIF. Please see Appendix A.3 for a detailed description.

## 5 EXPERIMENTS

To test the effectiveness of the proposed SSIF, we evaluate it on two challenging spatial-spectral super-resolution benchmark datasets – the CAVE dataset (Yasuma et al., 2010b) and the Pavia Centre dataset[7]. Both datasets are widely used for super-resolution tasks on hyperspectral images. Please refer to Appendix A.6 and A.7 for a description of both datasets and SSIF's model training details.

### 5.1 BASELINES AND SSIF MODEL VARIANTS

Compared with spatial SR and spectral SR, there has been much less work on SSSR. We mainly compare our model with 10 baselines[8]: **RCAN + AWAN**, **AWAN + RCAN**, **AWAN + SSPSR**, **RC/AW + MoG-DCN**, **SSJSR**, **US3RN**, **SSFIN**, **LIIF**, **CiaoSR**, and **LISSF**. Please refer to Appendix A.4 for a detailed description of each baseline. For the first 7 baselines, we have to train separate SR models for different spatial and spectral resolutions of the output images. LIIF and CiaoSR can use one model to generate output images with different spatial resolutions. However, we still need to train separate models for $\mathbf{I}^{hr-h}$ with different band numbers $C$. In contrast, $SSIF$ and LISSF can handle different spatial and spectral resolutions with one model.

---

[7] http://www.ehu.eus/ccwintco/index.php/Hyperspectral_Remote_Sensing_Scenes

[8] We do not pick LISSF as one baseline since it cannot handle RGB or multispectral images as input.

Based on the response functions we use (Gaussian or Uniform) and the wavelength sampling methods (Sampled or Fixed), we have 4 SSIF variants: **SSIF-RF-GS**, **SSIF-RF-GF**, **SSIF-RF-US**, and **SSIF-RF-UF**. We also consider 1 additional SSIF variant – **SSIF-M** which only use band middle point to represent each band. Please refer to Appendix A.5 for a detailed description of them.

## 5.2 SSSR ON THE CAVE DATASET

Table 1 shows the evaluation result of the SSSR task across different spatial scales $p$ on the original CAVE dataset with 31 bands. We use three evaluation metrics - PSNR, SSIM, and SAM which measure the quality of generated images from different perspectives. We evaluate different baselines as well as $SSIF$ under different spatial scales $p = \{2, 4, 8, 10, 12, 14\}$. We can see that:

1. All 4 SSIF-RF-* models can outperform or are comparable to the 10 baselines across all tested spatial scales even if the first 7 baselines are trained separately on each $p$.
2. SSIF-RF-GS achieves the best or 2nd best results across all spatial scales and metrics.
3. A general pattern we can see across all spatial scales is that the order of the model performances is SSIF-RF-* > CiaoSR > LIIF > LISSF and other 7 baselines. For more statistical significance analysis see the error bar plots shown in Figure 12 in Appendix A.8.2.

Table 1: Results for the image SSSR task across different spatial scales $p$ on the original CAVE (Yasuma et al., 2010a) dataset with 31 bands. "In-distribution" and "Out-of-distribution" indicate whether the model has seen this spatial scale $p$ during training. "Out-of-distribution" prediction is only applicable to LIIF (Chen et al., 2021), CiaoSR (Cao et al., 2023), LISSF (Zhang et al., 2024), and $SSIF$ models. The performance of these models across different $p$ are obtained from the same model while for other 7 baselines, we trained separated SR models for each spatial scale $p$. Except for LIIF, CiaoSR, and LISSF (Zhang et al., 2024), the performances of all the other 7 baselines are from (Ma et al., 2022)*. We highlight the best model for each setting in bold and underline the second-best model.

| Model | In-distribution | | | | | | | | |
|---|---|---|---|---|---|---|---|---|---|
| Spatial Scale $p$ | 2 | | | 4 | | | 8 | | |
| Metric | PSNR ↑ | SSIM ↑ | SAM ↓ | PSNR ↑ | SSIM ↑ | SAM ↓ | PSNR ↑ | SSIM ↑ | SAM ↓ |
| RCAN + AWAN(Ma et al., 2021)* | 36.22 | 0.971 | 8.81 | 32.69 | 0.935 | 9.82 | 28.25 | 0.834 | 11.73 |
| AWAN + RCAN(Ma et al., 2021)* | 36.09 | 0.969 | 8.42 | 31.44 | 0.916 | 9.24 | 27.77 | 0.837 | 12.39 |
| AWAN + SSPSR(Ma et al., 2021)* | 36.16 | 0.969 | 8.49 | 32.34 | 0.928 | 9.25 | 28.19 | 0.860 | 10.97 |
| RC/AW+MoG-DCN(Dong et al., 2021)* | 36.12 | 0.969 | 8.53 | 32.68 | 0.923 | 9.44 | 28.33 | 0.853 | 13.20 |
| SSJSR(Mei et al., 2020)* | 35.51 | 0.970 | 7.67 | 30.90 | 0.916 | 9.30 | 27.30 | 0.844 | 9.28 |
| US3RN(Ma et al., 2021)* | 36.18 | 0.972 | 7.43 | 32.90 | 0.942 | 7.91 | 28.81 | 0.887 | 9.02 |
| SSFIN(Ma et al., 2022)* | 37.36 | 0.977 | **6.49** | 33.41 | 0.947 | 7.11 | 29.21 | 0.896 | 8.07 |
| LIIF(Chen et al., 2021) | 36.82 | 0.977 | 6.85 | 34.36 | 0.956 | 7.31 | 31.26 | 0.900 | 8.32 |
| CiaoSR(Cao et al., 2023) | 37.09 | 0.977 | 6.85 | 34.57 | 0.954 | 9.36 | 32.05 | 0.913 | 7.84 |
| LISSF(Zhang et al., 2024) | 35.88 | 0.962 | 10.15 | 34.57 | 0.936 | 10.16 | 32.00 | 0.908 | 10.85 |
| SSIF-M | 36.08 | 0.952 | 10.22 | 34.45 | 0.937 | 10.32 | 32.27 | 0.901 | 10.78 |
| SSIF-RF-GS | **38.23** | **0.979** | 6.92 | **36.23** | **0.965** | 7.00 | **33.54** | **0.931** | 7.32 |
| SSIF-RF-GF | 37.42 | 0.977 | 7.09 | 35.47 | 0.963 | 7.09 | 32.98 | 0.928 | 7.68 |
| SSIF-RF-US | 37.98 | 0.977 | 6.66 | 35.65 | 0.963 | **6.90** | 33.21 | 0.930 | **7.29** |
| SSIF-RF-UF | 37.41 | 0.976 | 7.04 | 35.53 | 0.962 | 7.41 | 33.00 | 0.927 | 8.09 |
| Model | Out-of-distribution | | | | | | | | |
| Spatial Scale $p$ | 10 | | | 12 | | | 14 | | |
| Metric | PSNR ↑ | SSIM ↑ | SAM ↓ | PSNR ↑ | SSIM ↑ | SAM ↓ | PSNR ↑ | SSIM ↑ | SAM ↓ |
| LIIF(Chen et al., 2021) | 29.97 | 0.867 | 9.51 | 29.00 | 0.844 | 9.90 | 28.26 | 0.827 | 10.36 |
| CiaoSR(Cao et al., 2023) | 30.55 | 0.877 | 8.19 | 29.36 | 0.851 | 8.61 | 28.55 | 0.832 | 8.82 |
| LISSF(Zhang et al., 2024) | 31.06 | 0.875 | 11.27 | 30.18 | 0.858 | 11.40 | 29.67 | 0.845 | 11.51 |
| SSIF-M | 31.27 | 0.880 | 11.13 | 30.40 | 0.860 | 11.19 | 29.59 | 0.844 | 11.68 |
| SSIF-RF-GS | **32.20** | 0.909 | 7.87 | 31.14 | 0.891 | 8.19 | 30.44 | 0.878 | 8.57 |
| SSIF-RF-GF | 32.03 | 0.911 | 8.02 | 31.20 | 0.895 | 8.21 | 30.38 | 0.881 | 8.62 |
| SSIF-RF-US | 32.18 | **0.912** | **7.70** | **31.26** | **0.895** | **7.90** | **30.52** | **0.882** | **8.23** |
| SSIF-RF-UF | 31.82 | 0.906 | 8.57 | 30.83 | 0.887 | 8.86 | 30.19 | 0.874 | 9.14 |

Figure 3(a) and 3(b) compare model performances under different $C$ with a fixed spatial scale ($p = 4$ and $p = 8$ respectively). We can see that:

1. Both Figure 3(a) and 3(b) show that SSIF-RF-GS achieves the best performances in two spatial scales on both "in-distribution" and "out-of-distribution" spectral resolutions.
2. The performance of SSIF with fixed set of wavelengths during training (SSIF-RF-UF, SSIF-RF-GF, and SSIF-M) drop significantly when $C > 31$ while SSIF with randomized wavelengths (SSIF-RF-GS and SSIF-RF-US) generalized well for $C > 31$.
3. A general pattern can be observed – the order of model performance is SSIF-RF-* > CiaoSR > LIIF > LISSF >other 7 baselines.

## 5.3 SSSR ON THE PAVIA CENTRE REMOTE SENSING DATASET

Table 2 shows the evaluation results of the SSSR task across different spatial scales $p = \{2, 3, 4, 8, 10, 12, 14, 16\}$ on the original Pavia Centre dataset with 102 bands. We can see that:

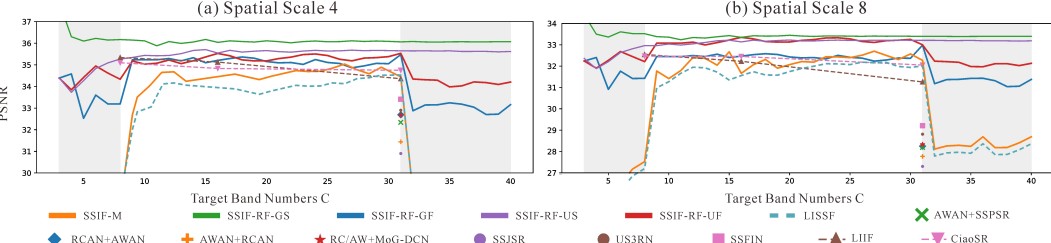

Figure 3: Results (PSNR) of different models on the SSSR task across different $C$ on the CAVE (Yasuma et al., 2010a) dataset. Here, the x axis indicates the number of bands $C$ of $\mathbf{I}^{hr-h}$. (a) and (b) compare the performances of different models across different $C$ in two spatial scales $p = 4$ and $p = 8$. Since our $SSIF$ can generalize to different $p$ and $C$, the evaluation metrics of each $SSIF$ are generated by one trained model. The gray area in these plots indicates "out-of-distribution" performance in which $SSIF$ are evaluated on $C$s which have not been used for training. Please see Figure 10 in Appendix A.8 for the evaluation results on three metrics.

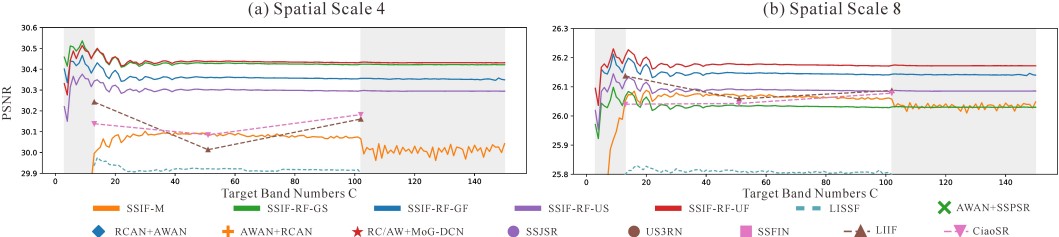

Figure 4: Evaluations across different $C$ on the Pavia Centre dataset. The setup is the same as Figure 3. Note that some of the baseline models do not appear in plots because the performances of them are very low and cannot be shown in the current metric range. Please see Figure 11 in Appendix A.8 for the results on three metrics.

1. All SSIF-RF-* can outperform all baselines on all spatial scales.
2. The performances of 4 SSIF-RF-* models are very similar across different spatial scales, and they outperform LISSF, CiaoSR, and LIIF in most settings.

Table 2: Image super-resolution on the original Pavia Centre (Yasuma et al., 2010a) dataset with 102 bands. We evaluate models across different spatial scales $p = \{2, 3, 4, 8, 10, 12, 14, 16\}$. "In-distribution" and "Out-of-distribution" have the same meaning as Table 1. The performance of LIIF, CiaoSR, LISSF, and SSIF across different $p$ are obtained from the same models while the other 7 baselines need to be trained separately on each $p$. Except for LIIF, CiaoSR, and LISSF, the performances of all the other 7 baselines are from Ma et al. (2022)*.

| Model | In-distribution | | | | | | | | | | | |
|---|---|---|---|---|---|---|---|---|---|---|---|---|
| Spatial Scale $p$ | 2 | | | 3 | | | 4 | | | 8 | | |
| Metric | PSNR ↑ | SSIM ↑ | SAM ↓ | PSNR ↑ | SSIM ↑ | SAM ↓ | PSNR ↑ | SSIM ↑ | SAM ↓ | PSNR ↑ | SSIM ↑ | SAM ↓ |
| RCAN + AWAN(Ma et al., 2021)* | 34.23 | 0.932 | 4.38 | 29.67 | 0.829 | 5.60 | 27.60 | 0.732 | 6.63 | 23.91 | 0.496 | 8.45 |
| AWAN + RCAN(Ma et al., 2021)* | 34.54 | 0.936 | 4.38 | 29.66 | 0.827 | 5.70 | 27.61 | 0.734 | 6.69 | 23.67 | 0.515 | 8.87 |
| AWAN + SSPSR(Ma et al., 2021)* | 34.24 | 0.934 | 4.30 | 29.60 | 0.828 | 5.55 | 27.71 | 0.742 | 6.32 | 24.21 | 0.506 | 8.14 |
| RC/AW+MoG-DCN(Dong et al., 2021)* | 34.01 | 0.929 | 4.91 | 29.77 | 0.833 | 5.53 | 27.59 | 0.734 | 6.66 | 23.92 | 0.528 | 8.44 |
| SSJSR(Mei et al., 2020)* | 31.80 | 0.894 | 4.80 | 29.05 | 0.810 | 6.14 | 27.06 | 0.703 | 6.93 | 20.61 | 0.347 | 18.30 |
| US3RN(Ma et al., 2021)* | 35.86 | 0.951 | 3.71 | 30.38 | 0.857 | 4.88 | 28.23 | 0.775 | 5.80 | 24.26 | 0.548 | 7.96 |
| SSFIN(Ma et al., 2022)* | 35.75 | 0.950 | **3.65** | 30.79 | 0.880 | 4.95 | 27.75 | 0.762 | 5.70 | 24.18 | 0.535 | 8.15 |
| LIIF(Chen et al., 2021) | 36.08 | 0.957 | 3.99 | 32.12 | 0.909 | 4.86 | 30.16 | 0.849 | 5.31 | 26.09 | 0.608 | 7.01 |
| CiaoSR(Cao et al., 2023) | 36.46 | 0.960 | 3.83 | 30.96 | 0.884 | 5.26 | 30.18 | 0.851 | 5.12 | 26.08 | 0.618 | 6.82 |
| LISSF(Zhang et al., 2024) | 35.79 | 0.954 | 4.55 | 30.17 | 0.875 | 5.17 | 29.88 | 0.825 | 5.79 | 25.12 | 0.598 | 7.12 |
| SSIF-M | 35.87 | 0.956 | 4.33 | 29.82 | 0.851 | 5.80 | 30.07 | 0.848 | 5.48 | 26.06 | 0.610 | 7.03 |
| SSIF-RF-GS | **36.84** | **0.962** | 3.71 | **32.31** | **0.910** | 4.61 | 30.42 | 0.858 | **4.99** | 26.03 | 0.619 | 6.77 |
| SSIF-RF-GF | 36.71 | 0.962 | 3.74 | 32.28 | 0.910 | 4.62 | 30.36 | 0.857 | 5.02 | 26.14 | 0.628 | 6.75 |
| SSIF-RF-US | 36.46 | 0.960 | 3.97 | 31.64 | 0.897 | 4.95 | 30.30 | 0.855 | 5.17 | 26.09 | 0.622 | 6.85 |
| SSIF-RF-UF | 36.79 | 0.962 | 3.73 | 32.27 | 0.909 | 4.64 | **30.43** | **0.858** | 5.00 | **26.17** | **0.629** | **6.71** |

| Model | Out-of-distribution | | | | | | | | | | | |
|---|---|---|---|---|---|---|---|---|---|---|---|---|
| Spatial Scale $p$ | 10 | | | 12 | | | 14 | | | 16 | | |
| Metric | PSNR ↑ | SSIM ↑ | SAM ↓ | PSNR ↑ | SSIM ↑ | SAM ↓ | PSNR ↑ | SSIM ↑ | SAM ↓ | PSNR ↑ | SSIM ↑ | SAM ↓ |
| LIIF(Chen et al., 2021) | 24.87 | 0.512 | 7.85 | 24.20 | 0.447 | 8.25 | **23.77** | 0.401 | 8.53 | **23.60** | 0.376 | 8.54 |
| CiaoSR(Cao et al., 2023) | 23.50 | 0.453 | 8.53 | 22.86 | 0.407 | 9.14 | 22.30 | 0.359 | 9.78 | 22.10 | 0.345 | 9.91 |
| LISSF(Zhang et al., 2024) | 24.58 | 0.505 | 8.12 | 23.64 | 0.451 | 8.59 | 23.44 | 0.373 | 8.88 | 23.41 | 0.377 | 8.84 |
| SSIF-M | 24.82 | 0.518 | 7.78 | 23.71 | 0.408 | 8.53 | 23.46 | 0.374 | 8.78 | 23.34 | 0.354 | 8.91 |
| SSIF-RF-GS | 24.86 | 0.523 | **7.52** | 24.05 | 0.443 | 8.05 | 23.66 | 0.401 | **8.37** | 23.52 | **0.382** | **8.50** |
| SSIF-RF-GF | 24.81 | 0.523 | 7.53 | **24.21** | **0.451** | **7.98** | 23.70 | **0.402** | **8.37** | 23.51 | 0.375 | **8.50** |
| SSIF-RF-US | **24.89** | **0.525** | 7.59 | 24.03 | 0.441 | 8.17 | 23.67 | 0.397 | 8.40 | 23.52 | 0.378 | 8.55 |
| SSIF-RF-UF | 24.88 | 0.521 | 7.53 | 24.15 | 0.447 | 8.02 | 23.65 | 0.400 | 8.40 | 23.44 | 0.373 | 8.58 |

Figure 4(a) and 4(b) compare different models across different spectral resolutions under two fixed spatial scales ($p = 4$ and 8 respectively). We can see that:

1. 4 SSIF-RF-* models can outperform all 10 baselines across different $C$ when $p = 4$. When $p = 8$, they outperform or are on the bar with CiaoSR and LIIF while outperforming other 8 baselines.
2. All 4 SSIF-RF-* show good generalization for "out-of-distribution" spectral scales, especially when $C > 102$ while SSIF-M suffers from performance degradation.

### 5.4 SPECTRAL SR, SPATIAL SR EXPERIMENTS AND ABLATION STUDIES

In addition to those 10 baselines, three specialized spectral SR models – HDNet (Hu et al., 2022), MST++ (Cai et al., 2022), and SSRNet (Dian et al., 2023) – were used for benchmarking on the spectral SR task using the CAVE and Pavia Centre datasets. The results, detailed in Appendix A.11, show that SSIF either outperforms or is on par with these task-specific baselines. Notably, SSIF also possesses the flexibility to handle both spatial and spectral SR simultaneously. We also compare CiaoSR and SSIF on spatial SR task. Results in Appendix A.12 show that SSIF can outperform or be on bar with CiaoSR even without the multiple spectral scale training process. Table 7 in Appendix A.13 compares the computational complexity of different models which shows that SSIF can achieve the SOTA performance without significantly increasing the model complexity.

Ablation studies on different designs of image encoder $E^I$, pixel feature decoder $F^\mathbf{x}$, and spectral decoder $D^{\mathbf{x},\lambda}$ on the CAVE dataset can be seen in Appendix A.9.1 and A.9.2. We find that using SwinIR as $E^I$, CiaoSR as $F^\mathbf{x}$, and dot product function as $D^{\mathbf{x},\lambda}$ leads to the best performance of SSIF. We also conduct an ablation study for $K$ on Pavia Centre dataset (see Figure 15 in Appendix A.9.3) and find out that a larger $K$ usually leads to better performance and better generalizability on unseen $C$. It shows that SSIF-RF-GF models with small $K$s also suffer from performance drop when $C > 102$ just like what we see in the CAVE experiments while bigger $K$s will mitigate this problem.

### 5.5 ANALYSIS

**Qualitative Results** In Figure 5, we provide qualitative comparisons of SSSR results from different methods. We can see that SSIF is much better at synthesizing sharp textures than other methods. Figure 6 shows the SSIF has superior performance on spectral reconstruction with extreme band numbers and significantly outperforms other methods. More results can be seen in Appendix A.16.

**What the Spectral Encoder Learned?** To understand how the spectral encoder represents a given wavelength $\lambda$ we plot each dimension of spectral embedding against $\lambda$ (Figure 7). We find that they generally resemble piecewise-linear PL basis functions (Paul & Koch, 1974) or the continuous PK basis functions (Melal, 1976). This makes sense because PL and PK are classical methods to represent a scalar function – i.e., $G^{\mathbf{x},\lambda}(\mathbf{S}^{lr-m}, \mathbf{x}, \cdot)$ in our case. We can think that the weights of these bases are provided by the $E^I$ and $F^\mathbf{x}$ given $\mathbf{I}^{lr-m}$ and $\mathbf{x}$. Having a spectral encoder with learnable parameters can potentially provide better representations than fixed basis functions.

**The Advantages of Physics-Inspired Design of SSIF** We find out that due to the incorporation of physical principles of spectral imaging in SSIF's model design, compared with other SIFs, SSIF is more data efficient, parameter efficient, and training efficient. Figure 8a shows that SSIF-RF-GS is more data efficient and can consistently outperform CiaoSR and SSIF-M across different training data sampling ratios. Figure 8b shows SSIF-RF-GS is more training efficient since it can converge faster. See Appendix A.10 for detailed explanations.

## 6 CONCLUSION

In this work, we propose Spatial-Spectral Implicit Function (SSIF), a physics-inspired neural implicit model that represents an image as a continuous function of both pixel coordinates in the spatial domain and wavelengths in the spectral domain. This enables SSIF to handle SSSR tasks with different output spatial and spectral resolutions simultaneously with one model. In contrast, all previous works have to train separate SR models for different spectral resolutions. We demonstrate the effectiveness of SSIF on the SSSR task with two datasets – CAVE and Pavia Centre. We show that SSIF can outperform all baselines across different spatial and spectral scales even when the baselines are allowed to be trained separately at each spectral resolution, thus solving an easier task. We demonstrate that SSIF generalizes well to unseen spatial and spectral resolutions. Moreover, we show that compared with other SIFs, due to its physics-inspired nature, SSIF is much more data efficient, parameter efficient, and training efficient.

In this study, the effectiveness of SSIF is mainly shown on hyperspectral image SR, while SSIF is flexible enough to handle multispectral images with irregular wavelength intervals. This will be studied in future work. Moreover, the data limitation of the hyperspectral images poses a significant

challenge to SR model training. We also plan to construct a large dataset for hyperspectral image SR. SSIF also has the risk of generating Deepfakes. Therefore, a holistic evaluation of SSIF on various downstream tasks is one of our future works.

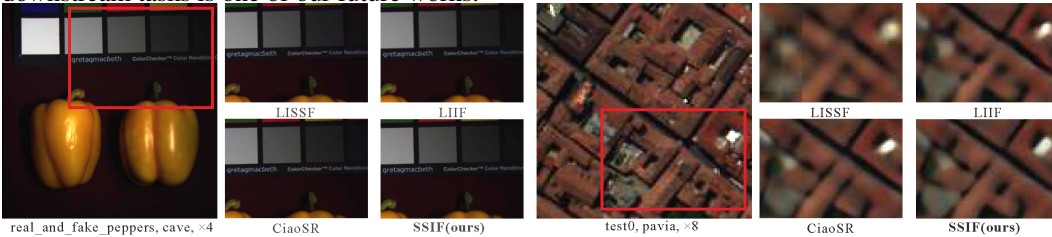

Figure 5: Visual comparison of spatial SR results using different methods on the CAVE (Yasuma et al., 2010a) (×4) and Pavia Centre dataset (×8). We zoom in the red box region from the ground truth image.

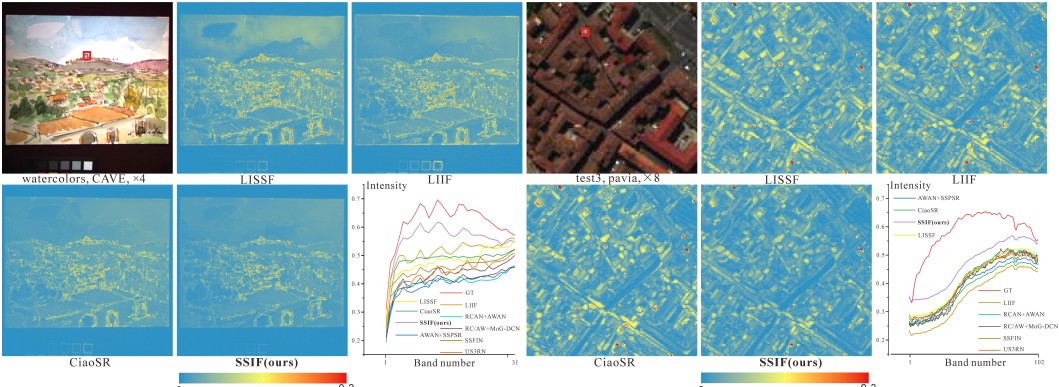

Figure 6: Visualization of the error maps of different methods of spectral reconstruction from MSI images on the CAVE (Yasuma et al., 2010a) (×4) and Pavia Centre dataset (×8). Mean Absolute Error across all reconstructed bands is used for error calculation. We also compare the reconstructed spectral signatures (spectral intensity) of selected pixels from different methods and mark them with red rectangles in the RGB image.

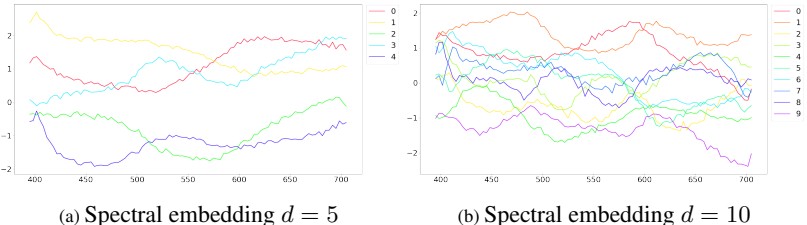

(a) Spectral embedding $d = 5$  (b) Spectral embedding $d = 10$

Figure 7: Visualizations of the spectral embeddings with small spectral embedding dimensions $d = \{5, 10\}$. Here we draw a curve for each dimension of the embedding, derived from the spectral encoders $E^\lambda$ of two learned SSIF-RF-GS. The x-axis indicates the wavelength and each curve $E^\lambda(\lambda)[j]$ corresponds to the values of a specific spectral embedding dimension $j$.

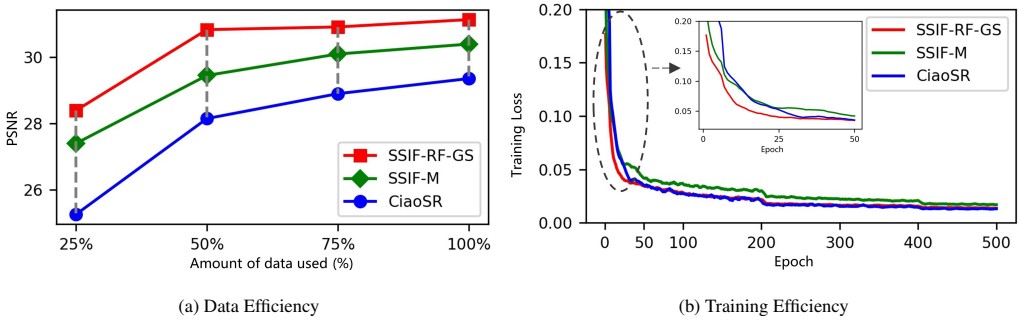

(a) Data Efficiency  (b) Training Efficiency

Figure 8: Evaluation results of data efficiency and training efficiency on SSSR task in CAVE dataset (Yasuma et al., 2010a). (a) We randomly sample 25%, 50% and 75% of the CAVE train set to train CiaoSR (Cao et al., 2023) and our SSIF model (SSIF-RF-GS, SSIF-M), respectively. Here we report the test result on spatial scale $p = 12$. (b) The training loss curve of three models in 500 epochs, gray circle indicates SSIF converges faster in the early training stage.

**Ethics Statement**    All datasets we use in this work including the CAVE and Pavia Centra datasets are publicly available datasets. Please refer to Appendix A.6 for a detailed description of both datasets. No human subject study is conducted in this work. We do not find specific negative societal impacts of this work. SSIF might have the risk of generating Deepfakes. A holistic evaluation of SSIF on various downstream tasks such as semantic segmentation and land use classification will be one of our future works.

**Reproducibility Statement**    Our source code has been uploaded as a supplementary file to reproduce our experimental results. The implementation details of the spectral encoder are described in Appendix A.2 and the dataset preparation details are discussed in Appendix A.3. All baselines used in the main experiments are described in Appendix A.4. The SSIF model training details are described in Appendix A.7.

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

# A APPENDIX

## A.1 A ILLUSTRATION OF USING SSIF FOR MULTITASK IMAGE SUPER-RESOLUTION

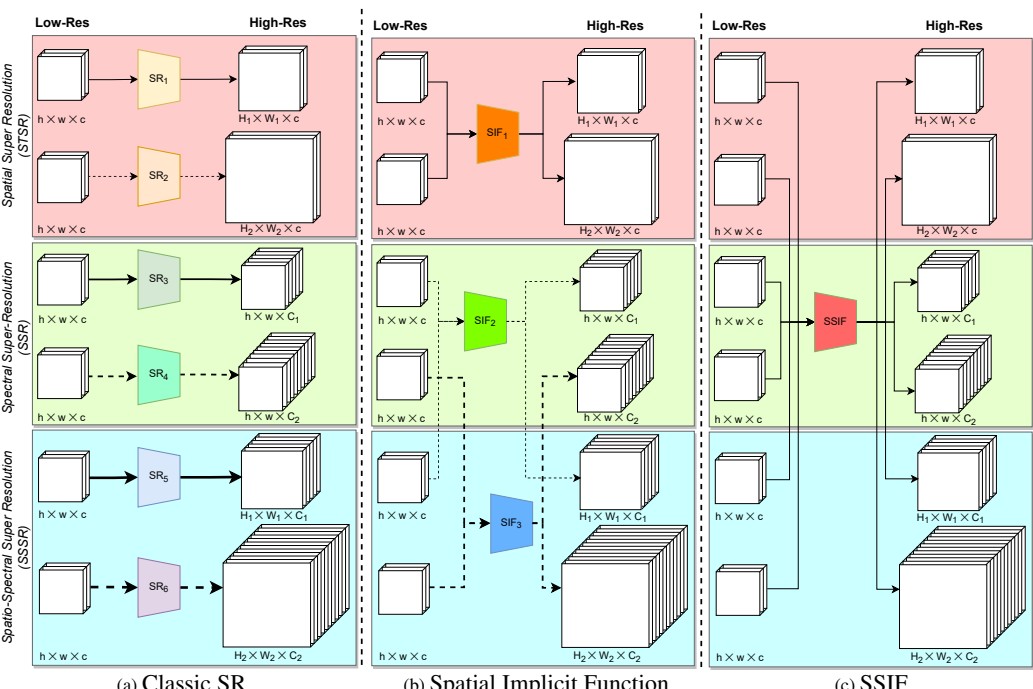

Figure 9: An illustration of image super-resolution on different spatial and spectral resolutions. The red, green, and blue boxes indicate three different super-resolution problems: Spatial Super-Resolution (spatial SR), Spectral Super-Resolution (spectral SR), and Spatio-Spectral Super-Resolution (SSSR). The three subfigures illustrate how the classic super-resolution models, the spatial implicit functions, and SSIF handle different SR tasks which generate images with different spatial and spectral resolutions. (a) Classic SR - most super-resolution models train **separate SR models** for different input and output image pairs with different spatial and spectral resolutions such as RCAN (Zhang et al., 2018a), SR3(Saharia et al., 2021), SSJSR (Mei et al., 2020), (He et al., 2021b), US3RN (Ma et al., 2021), SSFIN (Ma et al., 2022); (b) Spatial Implicit Function (SIF) - recently many research focused on using the idea of neural implicit function to develop spatial scale-agnostic super-resolution models such that one model can be trained to do super-resolution for different spatial scales such as MetaSR(Hu et al., 2019), LIIF(Chen et al., 2021), SADN (Wu et al., 2021a), ITSRN (Yang et al., 2021), (Zhang, 2021), and CiaoSR (Cao et al., 2023). However, they have to train separate SR models if target images have different spectral resolutions. (c) Spatial-Spectral Implicit Function ($SSIF$) aims at using one model to handle different spatial scales and spectral scales at the same time such that we can train one generic model for different SR tasks.

## A.2 SPECTRAL ENCODER $E^\lambda$

A key component of $SSIF$ is the spectral encoder $E^\lambda$ component. It consists of a Fourier feature mapping layer $\Psi(\cdot)$ followed by a multi-layer perceptron $MLP(\cdot)$:

$$\mathbf{b}_{i,k} = E^\lambda(\lambda_{i,k}) = MLP(\Psi(\lambda_{i,k})) \tag{8}$$

The Fourier feature mapping layer $\Psi(\cdot)$ takes a wavelength $\lambda_{i,k}$ sampled from the wavelength interval $\Lambda_i = [\lambda_{i,s}, \lambda_{i,e}] \in \Lambda^{hr-h}$ as the input and map it to a high dimensional vector $\mathbf{b}_{i,k} \in \mathbb{R}^d$, by using sinusoid functions with different frequencies. The idea is similar to the position encoder in Transformer (Vaswani et al., 2017), NeRF (Mildenhall et al., 2020), Space2Vec (Mai et al., 2020b; Tancik et al., 2020), and spatial implicit functions (Zhang, 2021; Dupont et al., 2021) for pixel location encoding. Here, we adopt the Space2Vec (Mai et al., 2020b) style position encoder $\Psi(\cdot)$. Let $\lambda_{min}, \lambda_{max}$ be the minimum and maximum scaling factor in the wavelength space, and $g = \frac{\lambda_{max}}{\lambda_{min}}$. We define $\Psi(\cdot)$ as Equation 9). Here, $\bigcup_{t=0}^{T-1}$ indicates vector concatenation through different scales.

$$\Psi(\lambda) = \bigcup_{t=0}^{T-1} \left[ \sin(\frac{\lambda}{\lambda_{min} \cdot g^{t/(T-1)}}), \cos(\frac{\lambda}{\lambda_{min} \cdot g^{t/(T-1)}}) \right]; \qquad (9)$$

### A.3 SUPER-RESOLUTION DATA PREPARATION

Figure 2a illustrates the data preparation process of SSIF. Given a training image pair which consists of a high spatial-spectral resolution image $\mathbf{I}_{max}^{hr-h} \in \mathbb{R}^{H \times W \times C_{max}}$ and an image with high spatial resolution but low spectral resolution $\mathbf{I}^{hr-m} \in \mathbb{R}^{H \times W \times c}$, we perform downsampling in both the spectral domain and spatial domain.

For the spectral downsampling process (the blue box in Figure 2a), we randomly sample a band number $C \sim Uni(C_{min}, C_{max})$ from a uniform distribution between the minimum and maximum band number $C_{min}, C_{max} > 0$. We use $C$ to downsample $\mathbf{I}_{max}^{hr-h}$ in the spectral domain which yield $\mathbf{I}^{hr-h} \in \mathbb{R}^{H \times W \times C}$. Then we convert $\mathbf{I}^{hr-h}$ into location-value-wavelength samples $(\mathbf{x}, \mathbf{s}^{hr-h}, \Lambda)$. $\mathbf{x}$ and $\Lambda$ serve as the input features while $\mathbf{s}^{hr-h}$ are the prediction target. Note that, here we can sample equally spaced wavelength intervals or irregular spaced wavelength intervals for the target HR-HSI images $\mathbf{I}^{hr-h}$ since SSIF is agnostic to this irregularity.

For the spatial downsampling (the orange box in Figure 2b), we randomly sample a spatial scale $p \sim Uni(p_{min}, p_{max})$ where $Uni(p_{min}, p_{max})$ is a uniform distribution between the minimum and maximum spatial scale $p_{min}, p_{max} > 0$. We use $p$ to spatially downsample $\mathbf{I}^{hr-m}$ into $\mathbf{I}^{lr-m} \in \mathbb{R}^{h \times w \times c}$ which serves as the input for $SSIF$. Here, $h = H/p$ and $w = W/p$.

Interestingly, when the spatial upsampling scale $p$ is fixed as 1, our SSIF is degraded to a spectral SR model. When the band $C$ is fixed as the same as the input band, i.e., $C = c$, SSIF is degraded to a spatial SR model. When we vary $C$ and $p$ during SSIF training, we allow the model to do spatial SR and spectral SR at different difficulty levels which helps it to learn a continuous representation both in the spatial and spectral domain.

### A.4 BASLINES

We consider 10 baselines in our SSSR task on two benchmark datasets:

1. **RCAN + AWAN** uses RCAN (Zhang et al., 2018a) for spatial SR and then AWAN (Li et al., 2020) for spectral SR in a sequential manner.

2. **AWAN + RCAN** simply reverses the order of RCAN and AWAN.

3. **AWAN + SSPSR** uses AWAN and SSPSR (Mei et al., 2020) for spectral SR and spatial SR.

4. **RC/AW + MoG-DCN** first separately uses RCAN (Zhang et al., 2018a) to do spatial SR to obtain HR-MSI images and uses AWAN (Li et al., 2020) to do spectral SR to obtain LR-HSI images, and then uses MoG-DCN (Dong et al., 2021) to do hyperspectral image fusion based on the previously generated HR-MSI and LR-HSI images.

5. **SSJSR** (Mei et al., 2020) uses a fully convolution-based deep neural network to do SSSR.

6. **US3RN** (Ma et al., 2021) uses a deep unfolding network to solve the SSSR problem with a closed-form solution.

7. **SSFIN** (Ma et al., 2022) follows the multi-task structure, first decoupling the SSSR into two tasks: spatial SR and spectral SR. Then it implements SSSR by feature fusion. It is the current state-of-the-art model for the SSSR task.

8. **LIIF** (Chen et al., 2021) is a spatial implicit function which was initially designed for spatial SR on multispectral data. We increase the output dimension of LIIF's final MLP to allow it to work on hyperspectral images.

9. **CiaoSR** (Cao et al., 2023) modifies the LIIF's nearest neighbor interpolation query feature into a self-attention-like architecture. We also change the output dimension of its final MLP to allow it to work on hyperspectral images.

10. **LISSF** (Zhang et al., 2024) is an implicit neural representation for joint SSSR of multispectral images in arbitrary scales. However, the input image encoder of LISSF utilizes 3D CNN layers, based on the assumption that the bands of the input images should have equal

spectral intervals between them, which is usually not the case in reality. In this paper, for a fair comparison, we replace its input image encoder backbone as SwinIR to be consistent with SSIF so that this modified LISSF can process input images with unequal band intervals.

### A.5 SSIF MODEL VARIANTS

We consider 4 SSIF variants: **SSIF-RF-GS**, **SSIF-RF-GF**, **SSIF-RF-US**, and **SSIF-RF-UF**. Both SSIF-RF-GS and SSIF-RF-GF uses a Gaussian distribution $\mathcal{N}(\mu_i, \sigma_i^2)$ as the response function for each band $b_i$ with wavelength interval $\Lambda_i = [\lambda_{i,s}, \lambda_{i,e}]$ where $\mu_i = \frac{\lambda_{i,s}+\lambda_{i,e}}{2}$ and $\sigma_i = \frac{\lambda_{i,e}-\lambda_{i,s}}{6}$. The difference is SSIF-RF-GS uses $\mathcal{N}(\mu_i, \sigma_i^2)$ to sample $K$ wavelengths from $\Lambda_i$ while SSIF-RF-GF uses fixed $K$ wavelengths with equal intervals in $\Lambda_i$. Similarly, both SSIF-RF-US and SSIF-RF-UF uses a Uniform distribution $\mathcal{U}(\lambda_{i,s}, \lambda_{i,e})$ as the response function for each band $b_i$. SSIF-RF-US uses $\mathcal{U}(\lambda_{i,s}, \lambda_{i,e})$ to sample $K$ wavelengths for each $\Lambda_i$ while SSIF-RF-UF uses fixed $K$ wavelengths with equal intervals.

We also consider 1 additional SSIF variant – **SSIF-M** which only uses band middle point $\mu_i = \frac{\lambda_{i,s}+\lambda_{i,e}}{2}$ for each wavelength interval, i.e., $K = 1$.

### A.6 DATASET DESCRIPTION

The CAVE dataset (Yasuma et al., 2010b) consists of 32 indoor hyperspectral (HSI) images captured under controlled illumination. Each image has a spatial size of $512 \times 512$ and 31 spectral bands covering the wavelength from 400nm to 700nm. Each HSI image is associated with an RGB image with the same spatial size. There are a lot of studies using the CAVE dataset for hyperspectral image super-resolution (Yao et al., 2020; Mei et al., 2020; Zhang et al., 2020c; Zhang, 2021; Han et al., 2021; Qu et al., 2021; Ma et al., 2021; 2022). However, these works focus on different SR tasks. In this work, we focus on the most challenging task – SSSR. The train/test split on the CAVE dataset varies from paper to paper. In order to keep a fair comparison to the previous study, we adopt the train/test split from SSFIN (Ma et al., 2022), the latest work on this dataset, and use the first 22 samples as the training dataset and the rest 10 samples as testing. The limited number of samples poses a significant challenge on modeling training. So similar to the previous work (Ma et al., 2021; Chen et al., 2021), given a HR-HSI and HR-MSI image pair $(\mathbf{I}_{max}^{hr-h}, \mathbf{I}^{hr-m})$, we first do random cropping for a $64p \times 64p$ image patch from both images. Then $\mathbf{I}^{hr-m}$ is spatially downsampled to a $64 \times 64$ image patch which serves as the input LR-MSI image $\mathbf{I}^{lr-m}$. We choose $p_{min} = 1$ and $p_{max} = 8$ for spatial downsampling, $C_{min} = 8$ and $C_{max} = 31$ for spectral downsampling (See Appendix A.3).

The Pavia Centre (PC) dataset is taken by ROSIS, a widely used hyperspectral sensor. The images were collected over the center area of Pavia, northern Italy, in 2001. It contains 102 spectral bands covering a spectrum from 430nm to 860nm. Figure 1a shows the spectral signature of one pixel A when $C = 102$. It has $1095 \times 715$ effective pixels. Similarly, we also adopt the train/test split from SSFIN (Ma et al., 2022) and crop the upper left $1024 \times 128$ pixels as the testing dataset and the rest for training. The PC dataset does not come with a multispectral image counterpart. So we adopt the practice of (Mei et al., 2020) to simulate the high-resolution multispectral (HR-MSI) image based on the sensor specification of the multispectral sensor of IKONOS. The resulted image has 4 bands which correspond to R, G, B, and NIR. Please see the MSI spectral signature in Figure 1a for reference. The same random cropping technique is used for PC. We choose $p_{min} = 1$ and $p_{max} = 8$ for spatial downsampling, $C_{min} = 13$ and $C_{max} = 102$ for spectral downsampling (See Appendix A.3).

### A.7 SSIF IMPLEMENTATION AND TRAINING DETAILS

We use SwinIR (Liang et al., 2021) as the image encoder $E^I$ and we use CiaoSR (Chen et al., 2021) as the pixel feature decoder $F^{\mathbf{x}}$. We ablate the combinations of different image encoders and pixel feature decoders in Figure 13 and we find the combination of SwinIR and CiaoSR performs the best.

For both the CAVE and Pavia Centre datasets, we first tune the learning rate $lr = \{5.e - 5, 1.e - 4, 2.e - 4\}$. We find out the default learning rate used by CiaoSR $lr = 1.e - 4$ works the best for both datasets.

Then we tune the hyperparameters of CiaoSR including the output image feature dimension for image encoder $E^I(\cdot) - d^I = \{64, 128, 256\}$, the input image size $h = w \in \{48, 64\}$, the hidden dimension of CiaoSR's multi-layer perceptron $- h_{LIIF} \in \{256, 512\}$. We find out $d^I = 64$, $h = w = 64$, and $h_{LIIF} = 256$ give us the best results of CiaoSR on CAVE while for the Pavia Centre, $d^I = 256$, $h = w = 64$, and $h_{LIIF} = 512$ yield the best results. In addition, we find out that using multiple PyTorch dataloaders with different input image sizes $h = w$ is especially useful for the Pavia Centre dataset. In our experiment, we use three different dataloaders with $\{16, 32, 64\}$ as their input image size.

After we get the best hyperparameter combination of CiaoSR, we directly use them for SSIF without tuning. And we only tune the newly added hyperparameters for SSIF including the hidden dimension $h_{SSIF} = \{512, 1024\}$ of $MLP(\cdot)$ in Equation 8 and the wavelength sampling number $K \in \{2, 4, 8, 16, 32, 48, 52, 64\}$. We find out $h_{SSIF} = 512$ and $K = 16$ are the best hyperparameter combination for the CAVE dataset and $h_{SSIF} = 1024$ and $K = 128$ is the best for the Pavia Centre dataset.

All experiments are conducted on a Linux server with 4 CUDA GPU of 24GB memory. We use the official implementations of all baselines[9]. We implement our SSIF in PyTorch and **the code is available in the supplementary file**. We will make SSIF's code publicly available upon acceptance.

### A.8 SUPPLEMENTARY EXPERIMENTAL RESULTS ON THE CAVE DATASET AND PAVIA CENTRE DATASET

#### A.8.1 SSSR MODEL COMPARISON ACROSS DIFFERENT SPECTRAL SCALES

While Figure 3 and 4 only show the comparison of different SSSR models' PSNR metrics on CAVE dataset and Pavia Centre dataset across different spectral scales, as their complementaries, Figure 10 and Figure 11 show the full plot of the comparison results of different SSSR models on all three metrics (i.e., PSNR, SSIM, and SAM) across different spectral scales on two datasets.

#### A.8.2 STATISTICAL SIGNIFICANCE ON OUR SSSR EXPERIMENTAL RESULTES

While Table 1 and 2 demonstrate the advantage of SSIF over all existing baselines on the SSSR task across all spatial scales, we do not report the statistical significance.

To show the robustness of the model, we compare our strongest baseline CiaoSR (Cao et al., 2023), and our SSIF-RF-GS model on the SSSR task across different spatial scales. More specifically, we retrain both models 3 times by using 3 different random seeds and plot their average performances as well as error bars, as shown in Fig. 12a and Fig. 12b. We can see that SSIF consistently outperforms CiaoSR across different spatial scales and different evaluation metrics, which proves the superiority of our approach.

### A.9 ABLATION STUDIES OF SSIF

#### A.9.1 ABLATION STUDIES OF SSIF'S $E^I$ AND $F^{\mathbf{x}}$ ON THE CAVE DATASET

We first study the impact of different image encoders $E^I$ and pixel feature decoders $F^{\mathbf{x}}$ on the performance of SSIF. Based on our best model on the CAVE dataset (i.e., SSIF-RF-GS), Table 3 and Figure 13 show the evaluation results of our ablation studies on (three) different image encoders $E^I$ and (two) pixel feature decoders $F^{\mathbf{x}}$ within SSIF model. We can see that:

1. Across all scales (both spatial and spectral), with different image encoder $E^I$, the performances of SSIF show a consistent pattern: SwinIR (Liang et al., 2021) > RDN (Zhang et al., 2018b) > EDSR(Lim et al., 2017).

---

[9]The LIIF and CiaoSR implementation is under BSD 3-Clause "New" or "Revised" License.

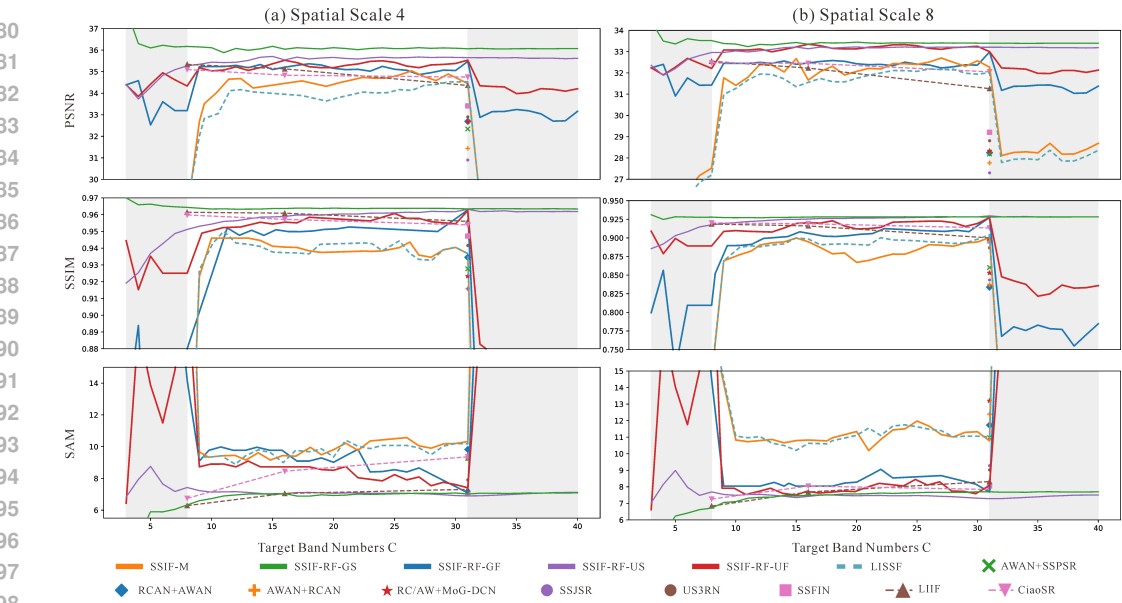

Figure 10: The evaluation results of various models on the SSSR task across different $C$ on the CAVE (Yasuma et al., 2010a) dataset. We compare their performances on three metrics: PSNR, SSIM, and SAM.

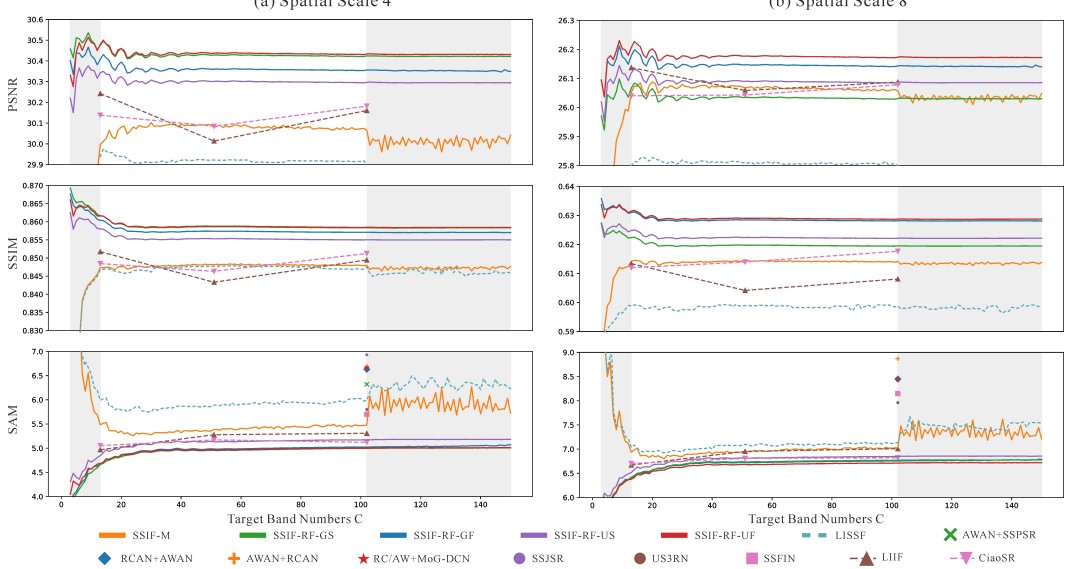

Figure 11: The evaluation results of various models on the SSSR task across different $C$ on the Pavia Centre dataset. We compare their performances on three metrics: PSNR, SSIM, and SAM.

2. Similarly, across all scales (both spatial and spectral), with different pixel feature decoder $F^{\mathbf{x}}$, the performances of SSIF also show a consistent pattern: CiaoSR (Cao et al., 2023) > LIIF Chen et al. (2021).

3. Using SwinIR as $E^I$ and CiaoSR as $F^{\mathbf{x}}$ in SSIF yields the best performance of SSIF.

4. The observed performance improvements are mutually beneficial. Employing a stronger combination of image encoder $E^I$ and pixel feature decoder $F^{\mathbf{x}}$ consistently boosts SSIF performance.

### A.9.2 ABLATION STUDIES OF SSIF'S $D^{\mathbf{x},\lambda}$ ON THE CAVE DATASET

Next, we conduct ablation studies of different designs of spectral decoder $D^{\mathbf{x},\lambda}$ on the CAVE dataset. The results are shown in Figure 14. One SSIF variant – SSIF-RF-GS is used here. We test three spectral decoder $D^{\mathbf{x},\lambda}$ variants:

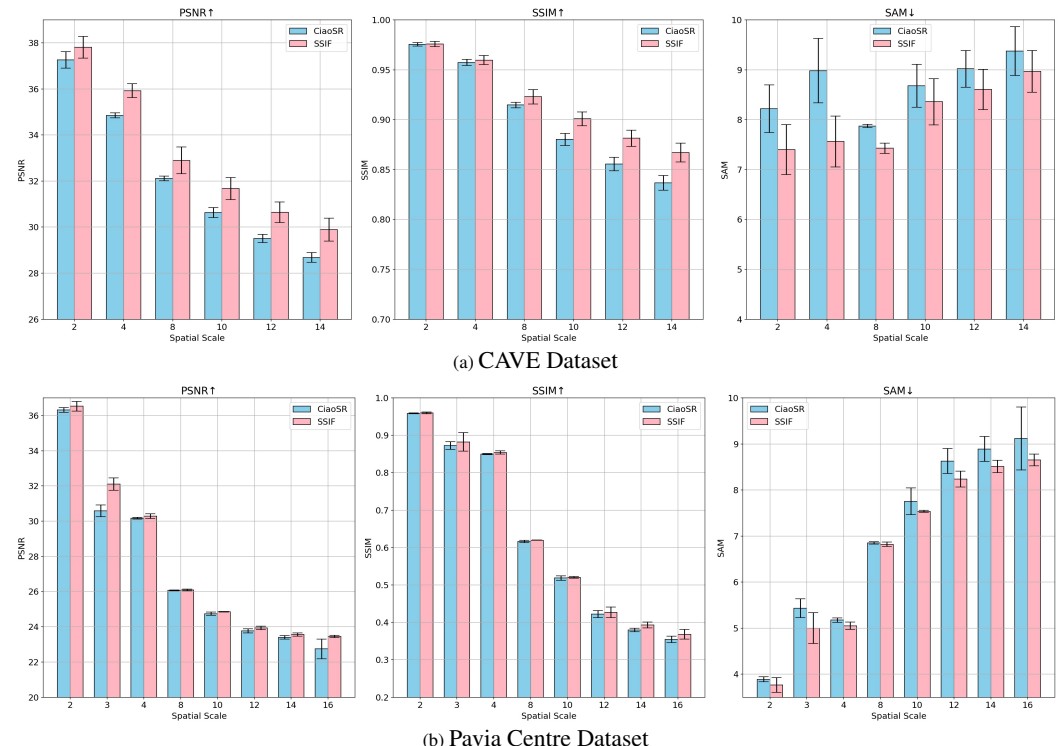

(a) CAVE Dataset

(b) Pavia Centre Dataset

Figure 12: The error bar of the SSSR performances of CiaoSR (Cao et al., 2023) and our SSIF-RF-GS on the (a) CAVE dataset (Yasuma et al., 2010a) and (b) Pavia Centre dataset. We use 3 different random seeds to retrain both models to obtain the results.

Table 3: Evaluation results of the ablation study on the impact of different image encoders $E^I$ (i.e., EDSR (Lim et al., 2017), RDN (Zhang et al., 2018b), and SwinIR (Liang et al., 2021)) and pixel feature decoders $F^{\mathbf{x}}$ (LIIF (Chen et al., 2021) and CiaoSR (Cao et al., 2023)) within our SSIF model for SSSR tasks across different spatial scales $p$ on the CAVE dataset with 31 bands.

| Ablations on image encoder & pixel feature decoder | | | In-distribution | | | | | | | | |
|---|---|---|---|---|---|---|---|---|---|---|---|
| Spatial Scale $p$ | | | 2 | | | 4 | | | 8 | | |
| Model | Image Encoder | Pixel Feature Decoder | PSNR↑ | SSIM↑ | SAM↓ | PSNR↑ | SSIM↑ | SAM↓ | PSNR↑ | SSIM↑ | SAM↓ |
| SSIF | EDSR (Lim et al., 2017) | LIIF (Chen et al., 2021) | 36.54 | 0.974 | 7.26 | 34.44 | 0.954 | 7.54 | 32.19 | 0.912 | 7.94 |
| | | CiaoSR (Cao et al., 2023) | 37.05 | 0.974 | 7.45 | 34.76 | 0.954 | 7.78 | 32.33 | 0.913 | 8.25 |
| SSIF | RDN (Zhang et al., 2018b) | LIIF (Chen et al., 2021) | 36.95 | 0.974 | 7.32 | 34.86 | 0.945 | 7.61 | 32.45 | 0.918 | 8.10 |
| | | CiaoSR (Cao et al., 2023) | 37.08 | 0.974 | 7.49 | 34.91 | 0.955 | 8.08 | 32.50 | 0.915 | 8.76 |
| SSIF | SwinIR (Liang et al., 2021) | LIIF (Chen et al., 2021) | 37.86 | 0.978 | 7.22 | 35.76 | 0.963 | 7.49 | 33.17 | 0.911 | 7.90 |
| | | CiaoSR (Cao et al., 2023) | **38.23** | **0.979** | **6.92** | **36.23** | **0.965** | **7.00** | **33.54** | **0.931** | **7.32** |
| Ablations on image encoder & pixel feature decoder | | | Out-of-distribution | | | | | | | | |
| Spatial Scale $p$ | | | 10 | | | 12 | | | 14 | | |
| Model | Image Encoder | Pixel Feature Decoder | PSNR↑ | SSIM↑ | SAM↓ | PSNR↑ | SSIM↑ | SAM↓ | PSNR↑ | SSIM↑ | SAM↓ |
| SSIF | EDSR (Lim et al., 2017) | LIIF (Chen et al., 2021) | 31.13 | 0.893 | 8.22 | 30.16 | 0.875 | 8.45 | 29.49 | 0.863 | 8.70 |
| | | CiaoSR (Cao et al., 2023) | 31.41 | 0.896 | 8.53 | 30.45 | 0.878 | 8.70 | 29.60 | 0.864 | 8.96 |
| SSIF | RDN (Zhang et al., 2018b) | LIIF (Chen et al., 2021) | 31.40 | 0.899 | 8.44 | 30.57 | 0.881 | 8.75 | 29.71 | 0.866 | 9.03 |
| | | CiaoSR (Cao et al., 2023) | 31.51 | 0.897 | 8.84 | 30.65 | 0.881 | 9.16 | 29.83 | 0.868 | 9.16 |
| SSIF | SwinIR (Liang et al., 2021) | LIIF (Chen et al., 2021) | 31.21 | 0.896 | 8.76 | 30.28 | 0.877 | 8.96 | 29.54 | 0.860 | 9.43 |
| | | CiaoSR (Cao et al., 2023) | **32.20** | **0.909** | **7.87** | **31.14** | **0.891** | **8.19** | **30.44** | **0.878** | **8.57** |

1. "**D**": $D^{\mathbf{x},\lambda}$ is a multilayer perceptron (MLP) which is modulated by the image feature embedding $\mathbf{h_x}$. $D^{\mathbf{x},\lambda}$ takes a spectral embedding $\mathbf{b}_{i,k}$ as the input and output the corresponding radiance value. When $D^{\mathbf{x},\lambda}$ is a one-layer MLP, this can be seen as the dot product between the input spectral embedding $\mathbf{b}_{i,k}$ and image feature embedding $\mathbf{h_x}$.

2. "**C**": $D^{\mathbf{x},\lambda}$ is a multilayer perceptron (MLP) which takes the concatenation of spectral embedding $\mathbf{b}_{i,k}$ and image feature embedding $\mathbf{h_x}$ as the input and output the corresponding radiance value.

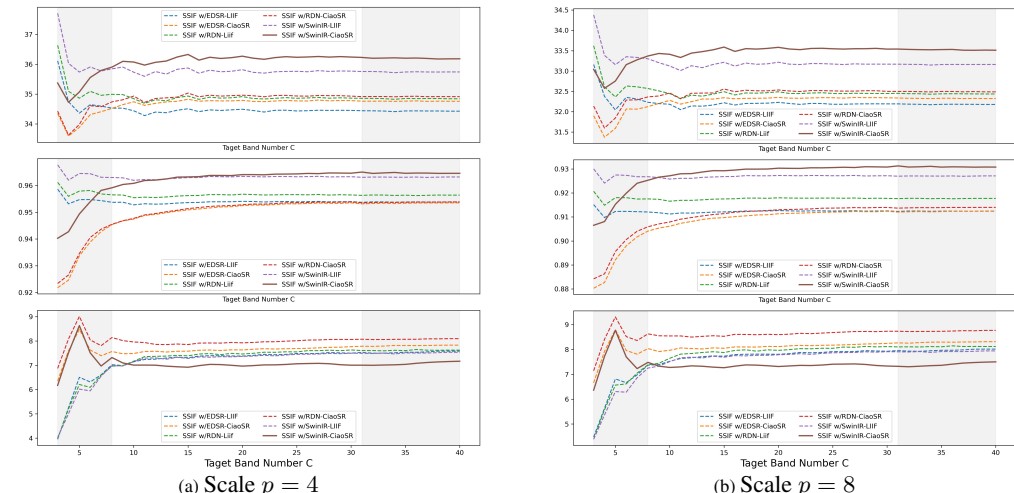

(a) Scale $p = 4$      (b) Scale $p = 8$

Figure 13: Evaluation results of the ablation study on the impact of different image encoders $E^I$ and pixel feature decoders $F^{\mathbf{x}}$ within our SSIF model for SSSR task across different band numbers $C$ on the CAVE dataset.

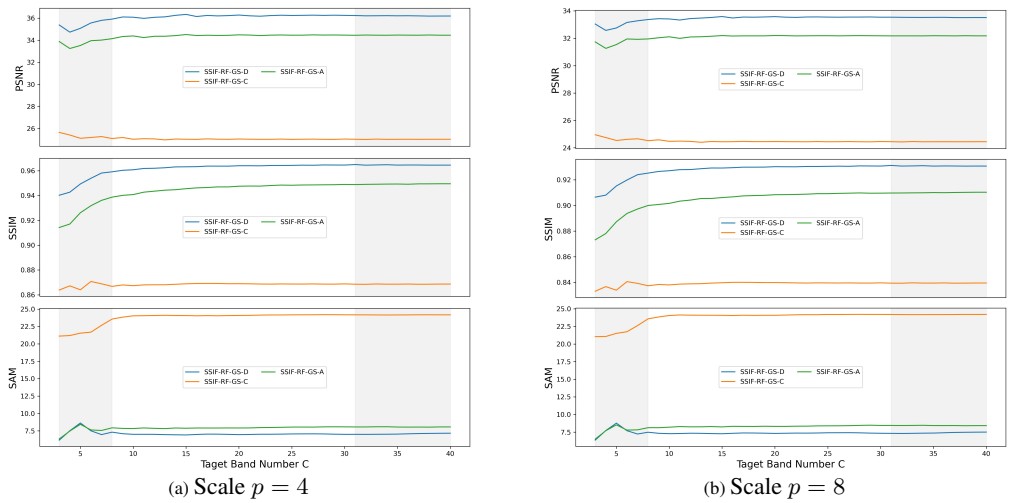

(a) Scale $p = 4$      (b) Scale $p = 8$

Figure 14: The ablation studies of different designs of spectral decoder $D^{\mathbf{x},\lambda}$ on the CAVE dataset. Here, we use one SSIF model – SSIF-RF-GS. Three spectral decoder $D^{\mathbf{x},\lambda}$ variants are explored: "**D**" "**C**" and "**A**". Gray areas indicate out-of-distribution spectral scales which have not been seen during SSIF training.

3. "**A**": $D^{\mathbf{x},\lambda}$ is a self-attention (Vaswani et al., 2017) based mechanism. It initially computes the dot product between the spectral embedding $\mathbf{b}_{i,k}$ and the image feature embedding $\mathbf{h}_{\mathbf{x}}$, then it applies self-attention function to re-weight the spectral and spatial information within the output embedding.

Three spectral decoders $D^{\mathbf{x},\lambda}$ amount to 3 different SSIF variants. From Figure 14, we can see that SSIF-RF-GS-D outperforms SSIF-RF-GS-A and SSIF-RF-GS-C across different spatial and spectral scales (on both in-distribution and out-of-distribution spectral scales) on all three metrics, which indicates that spectral decoder $D^{\mathbf{x},\lambda}$ variant **D** is usually more effective than **A** and **C**.

### A.9.3 ABLATION STUDIES OF THE NUMBER OF SAMPLED WAVELENGTHS ON THE PAVIA CENTRE DATASET

We conduct the ablation study on the effect of the number of sampled wavelengths in each wavelength interval $\Lambda_i - K$ on the model performance. We use the Pavia Centra dataset as an example and compare model performances of SSIF-RF-GF with different $K$. Figure 15 illustrates the results. We

can see that a bigger $K$ leads to better model performance and better generalizability on unseen spectral scales $C$. In other words, the performance of SSIF-RF-GF with larger $K$ is better across different $C$ and is more stable when $C > 102$.

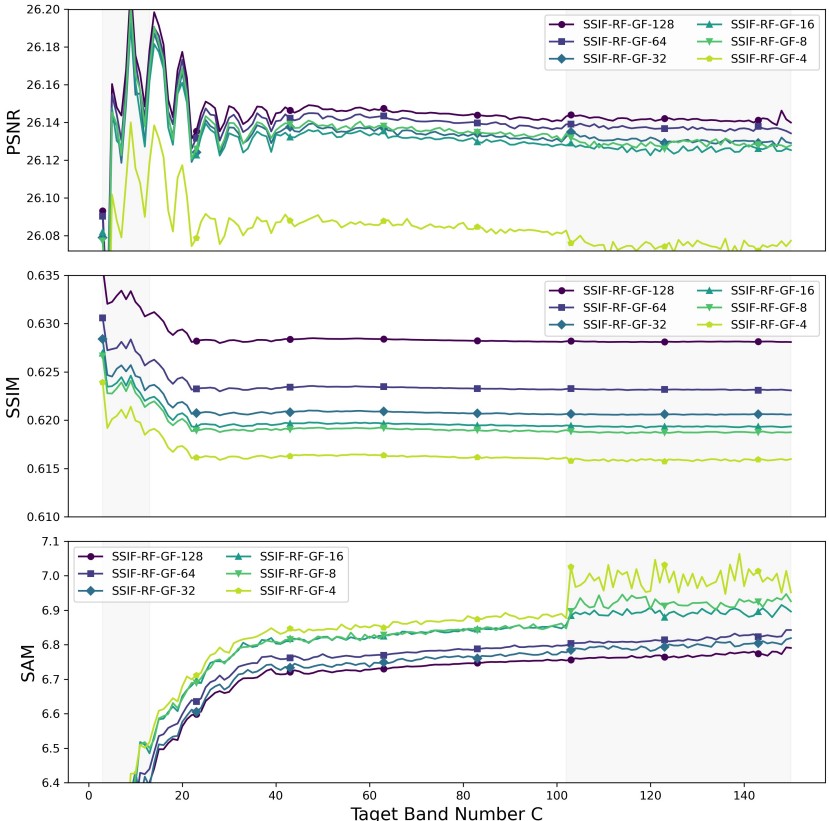

Figure 15: The ablation study on the number of sampled wavelengths in each wavelength interval $\Lambda_i - K$. We use the Pavia Centra dataset with spatial scale $p = 8$ as an example. The setting is similar to Figure 4. We use SSIF-RF-GF model as an example and tune the hyperparameter $K = \{4, 8, 16, 32, 64, 128\}$. Here, each SSIF is named as SSIF-RF-GF-K. We can see that a bigger $K$ leads to better model performance and better generalizability on unseen spectral scales (i.e., $C > 102$).

### A.10 Studies on the Advantages of Physics-Inspired Nature of SSIF

Compared with existing SIF models such as LIIF (Chen et al., 2021) and CiaoSR (Cao et al., 2023), SSIF has one big difference – it incorporates the physical principles of spectral imaging into the neural implicit function model design. We hypothesize that the physics-inspired nature of SSIF can lead to three advantages:

1. **Data efficiency**: Compared with other SIF models, SSIF will require less training data to achieve the same level of model performance. In other words, when trained with different proportions of training data, SSIF will always outperform other SIF models.

2. **Parameter efficiency**: Compared with other SIF models, SSIF requires a much smaller number of learnable parameters to achieve the same set of tasks.

3. **Training efficiency**: During model training, SSIF converges faster than other SIF models.

To validate our hypotheses, we conduct a series of experiments on the CAVE dataset (Yasuma et al., 2010a) by comparing our strongest baseline CiaoSR (Cao et al., 2023) with our SSIF-RF-* and SSIF-M.

We summarize our findings in Figure 16, Table 4, and Figure 17. We can see that:

1. SSIF-RF-* is indeed more data efficient than CiaoSR and SSIF's simple variant, SSIF-M, as shown in Figure 16. Since SSIF explicitly embeds the physical principles of spectral imaging into its model design, SSIF is less data-dependent and more robust. When trained with different proportions of training data, SSIF-RF-* can consistently outperform CiaoSR. In particular, SSIF shows great performance gains when trained with only 25% of the train set (at least 3.14 PSNR gain). Moreover, SSIF-RF-* can also consistently outperform SSIF-M which indicates that simply performing spectral encoding without considering the nature of sensors' response functions (as SSIF-M does) will lead to significant model performance degradation.

2. SSIF is also parameter efficient. It has similar numbers of learnable parameters as CiaoSR (see Table 4). However, we need to train separate CiaoSR models for different target spectral resolutions while one SSIF can handle all these tasks simultaneously.

3. SSIF is also training efficient as shown in Figure 17. As discussed above, since SSIF explicitly embeds the physics principles, it can converge faster as a result. This phenomenon is particularly evident in early epochs, as shown in Figure 17.

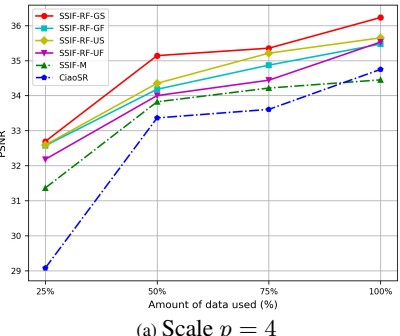 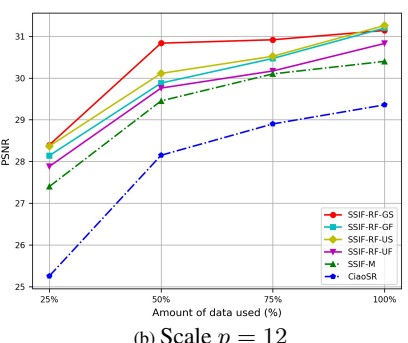

(a) Scale $p = 4$                    (b) Scale $p = 12$

Figure 16: Experiments to demonstrate the data efficiency of SSIF on spatial SR task with two spatial scales $p = 4$ and $p = 12$. We randomly sample 25%, 50% and 75% of the CAVE train set and use the sampled subsets to train CiaoSR (Cao et al., 2023) and our SSIF variants, i.e., SSIF-RF-* and SSIF-M. It is obvious that SSIF-RF-* consistently outperforms CiaoSR (Cao et al., 2023) and SSIF-M across different training data ratios.

Table 4: A comparison between SSIF and CiaoSR in terms of model parameters. We can see that SSIF is parameter efficient since with 0.3M additional parameters, it can simultaneously generate output images with various spectral resolutions while we have to train separate CiaoSR models for different spectral resolutions.

| Model | Model Size (MB) | Million Parameters |
|---|---|---|
| CiaoSR (Cao et al., 2023) | 169 | 13.0 |
| SSIF | 172 | 13.3 |

## A.11 SPECTRAL SR ON THE CAVE AND PAVIA CENTRE REMOTE SENSING DATASET

We evaluate the performance of SSIF on the single-image spectral SR task (i.e., keeping the spatial resolution unchanged while increasing the spectral resolution) and compare it with multiple baselines. In addition to the existing baselines, we also add three recent spectral SR models – HDNet (Hu et al., 2022), MST++ (Cai et al., 2022), and SSRNet (Dian et al., 2023). The spectral SR evaluation results on both datasets are summarized in Table 5, we can see that:

1. Three spectral SR baselines perform better than other baselines on both datasets, while four SSIF variants show competitive performance on both datasets.

2. On the CAVE dataset, SSIF-RF-GS outperforms all baselines for PSNR and SSIM, while remaining on par with 3 new baselines for SAM.

3. On the Pavia Centre dataset, SSIF-RF-US can outperform all baselines for PSNR and SSIM while being competitive for SAM. SSIF-RF-GS achieves the best SAM score.

HDNet (Hu et al., 2022), MST++ (Hu et al., 2022), and SSRNet (Hu et al., 2022) are specifically designed for spectral SR tasks. We have to train separate models for different spectral scales in

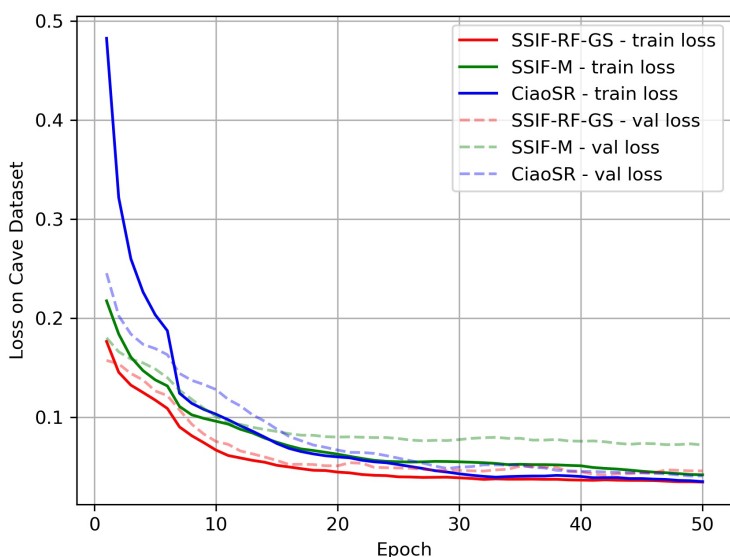

Figure 17: A comparison of training and validation loss curves for SSIF-RF-GS, SSIF-M and CiaoSR in the first 50 epochs. We can see that SSIF-RF-GS converges faster.

Table 5: The evaluation result of the spectral super-resolution task on CAVE(Yasuma et al., 2010a) and PAVIA Centra datasets. On the CAVE and PAVIA Centra datasets, we use RGB images and 4-band images as the respective input and benchmark model performance to reconstruct all hyperspectral bands – 31 and 102 bands respectively. Note that HDNet (Hu et al., 2022), MST++ (Cai et al., 2022), and SSRNet (Dian et al., 2023) are SOTA methods exclusively designed for spectral SR tasks, while our SSIF can tackle spatial SR, spectral SR, and SSSR tasks in arbitrary scales. All methods except for LISSF are implemented using their respective official codes, with hyperparameters selected from their respective papers.

| Method | CAVE Dataset | | | PAVIA Dataset | | |
|---|---|---|---|---|---|---|
| | PSNR↑ | SSIM↑ | SAM↓ | PSNR↑ | SSIM↑ | SAM↓ |
| RCAN + AWAN(Ma et al., 2021) | 39.1725 | 0.9745 | 7.5411 | 37.1532 | 0.9412 | 3.6554 |
| AWAN + RCAN(Ma et al., 2021) | 39.4221 | 0.9784 | 7.4742 | 37.2122 | 0.9401 | 3.7555 |
| AWAN + SSPSR(Ma et al., 2021) | 39.6511 | 0.9799 | 7.3155 | 37.0145 | 0.9544 | 4.0819 |
| RC/AW + MoG-DCN(Dong et al., 2021) | 39.2411 | 0.9723 | 7.4131 | 36.9283 | 0.9273 | 4.1211 |
| US3RN(Ma et al., 2021) | 40.1445 | 0.9801 | 7.0136 | 37.9338 | 0.9608 | 3.8764 |
| SSFIN(Ma et al., 2022) | 40.7596 | 0.9812 | 6.9713 | 38.0258 | 0.9721 | 3.614 |
| HDNet(Hu et al., 2022) | 42.9673 | 0.9809 | 6.7478 | 40.7674 | 0.9551 | 3.4914 |
| MST++(Cai et al., 2022) | 43.4765 | 0.9811 | **6.4257** | 40.8456 | 0.9549 | 3.4712 |
| SSRNet(Dian et al., 2023) | 43.3197 | 0.9800 | 6.7734 | 40.9170 | 0.9578 | 3.4653 |
| LIIF(Chen et al., 2021) | 41.4132 | 0.9738 | 6.9426 | 38.2002 | 0.9692 | 3.6811 |
| CiaoSR(Cao et al., 2023) | 41.5314 | 0.9771 | 6.9422 | 38.3148 | 0.9700 | 3.6628 |
| LISSF (Zhang et al., 2024) | 37.7852 | 0.9511 | 7.7781 | 35.1242 | 0.9511 | 4.5371 |
| SSIF-M | 38.2954 | 0.9682 | 7.4577 | 36.9544 | 0.9588 | 4.5210 |
| SSIF-RF-GS | **44.0124** | **0.9814** | 6.7324 | 40.4987 | 0.9812 | **3.4400** |
| SSIF-RF-GF | 43.5187 | 0.9783 | 6.8544 | 40.1455 | 0.9785 | 3.6515 |
| SSIF-RF-US | 43.2655 | 0.9788 | 6.8688 | **40.9563** | **0.9888** | 3.5710 |
| SSIF-RF-UF | 40.8507 | 0.9715 | 7.2901 | 40.8701 | 0.9864 | 3.5400 |

spectral SR. In contrast, SSIF just needs to be trained once to tackle spatial SR, spectral SR, and SSSR tasks in arbitrary spatial and spectral scales. SSIF outperforms or is on par with these three task-specific and scale-specific models, showing the superiority of SSIF.

## A.12 SPATIAL SR ON THE CAVE DATASET

When the spectral scale C/c = 1, the SSSR task degrades to the normal spatial SR task. The results are shown in Table 6. In order to make a fair comparison, both CiaoSR and SSIF have the same image encoder – SwinIR and the same pixel feature decoder – CiaoSR. They are trained with a fixed spectral

scale C/c = 1. From Table 6, we can see that SSIF outperforms CiaoSR across different spatial scales on different evaluation metrics. The only exceptions are PSNR and SAM when p = 2 and PSNR when p = 10. In these cases, SSIF also shows competitive performances. This demonstrates the advantages of SSIF over CiaoSR on the architecture side even without the multiple spectral scale training process.

Table 6: Evaluations of CiaoSR(Cao et al., 2023) and SSIF for the spatial SR task on CAVE dataset (Yasuma et al., 2010a). Here, we fix the spectral scale as 1 during SSIF training to make a fair comparison with CiaoSR.

| | In-distribution | | | | | | Out-of-distribution | | | | | |
|---|---|---|---|---|---|---|---|---|---|---|---|---|
| Spatial scale | 2 | | | 8 | | | 10 | | | 14 | | |
| Metrics | PSNR↑ | SSIM↑ | SAM↓ | PSNR↑ | SSIM↑ | SAM↓ | PSNR↑ | SSIM↑ | SAM↓ | PSNR↑ | SSIM↑ | SAM↓ |
| CiaoSR | **40.7741** | 0.9718 | **6.4201** | 35.1210 | 0.9401 | 7.0751 | **33.5548** | 0.9232 | 7.3800 | 28.6452 | 0.8815 | **8.9752** |
| SSIF (SwinIR-CiaoSR) | 40.7454 | **0.9794** | 6.0511 | **36.9974** | **0.9642** | **7.0024** | 33.5145 | **0.9313** | **7.2954** | **29.3421** | **0.8932** | 9.0125 |

## A.13 COMPARISON ON MODEL COMPUTATIONAL COMPLEXITY

As shown in Table 7, we compared SSIF with all baselines on model computational complexity with three metrics: the number of parameters (Params), FLOPS, and model size. We can see that SSIF's Param. and model size are comparable to many INR baselines such as LIIF, CiaoSR, and LISSF while they are much less than some CNN-based baselines such as AWAN+SSPSR and RC/AW+MoG-DCN. In terms of FLOPS, SSIF is slightly higher but it is comparable to CiaoSR which is the most similar model of SSIF. We can see that SSIF can achieve SOTA performance on the SSSR task without significantly increasing the model complexity.

Table 7: A comparison across different SSSR models on computational complexity. Since SSIF can use different image encoders and pixel feature decoders, SSIF (SwinIR-CiaoSR) indicates the version with the highest computational complexity – the one using CiaoSR as the image encoder and SwinIR as the pixel feature decoder.

| Methods | Params(M) | FLOPS(G) | Model Size (Mb) |
|---|---|---|---|
| RCAN + AWAN(Ma et al., 2021) | 17.3 | 40.3 | 201.4 |
| AWAN + RCAN(Ma et al., 2021) | 17.6 | 35.1 | 203.0 |
| AWAN + SSPSR(Ma et al., 2021) | 30.7 | 58.1 | 325.1 |
| RC/AW+MoG-DCN(Dong et al., 2021) | 434.3 | 491.2 | 499.3 |
| US3RN(Ma et al., 2021) | 2.7 | 64.2 | 30.7 |
| SSFIN(Ma et al., 2022) | 5.7 | 75.3 | 65.8 |
| LIIF(Chen et al., 2021) | 12.0 | 419.3 | 156.8 |
| CiaoSR(Cao et al., 2023) | 13.0 | 636.5 | 169.0 |
| LISSF (Zhang et al., 2024) | 13.1 | 725.4 | 174.6 |
| SSIF (SwinIR-CiaoSR) | 13.3 | 717.0 | 172.0 |

## A.14 DISCUSSIONS ON THE CHOICE OF $C_{min}$

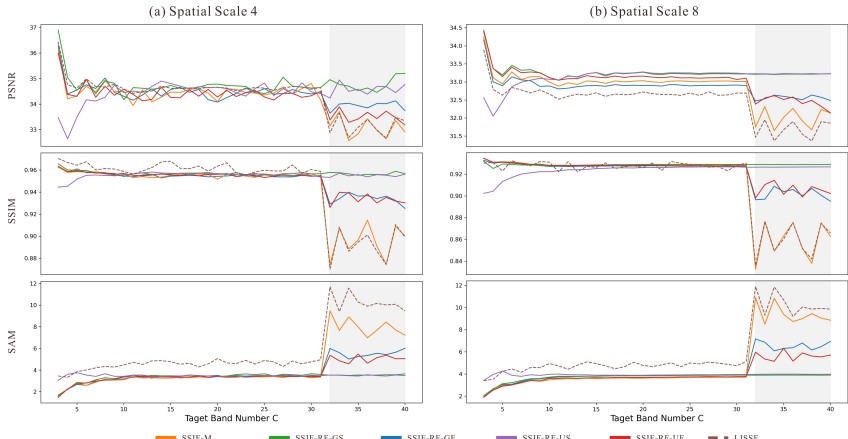

Figure 18: Evaluation results at different $C_{min}$ in the CAVE dataset. Instead of setting $C_{min} = 8$ as shown in Figure 3, we set $C_{min} = 1$ and retrain different SR models. The gray area indicates the area of out-of-distribution spectra.

To verify the model's consistent performance when facing with different number of $C$ in model training, instead of setting $C_{min} = 8$ as shown in Figure 3, we set the $C_{min} = 1$ and $C_{max} = 31$ when training the model, and then evaluate the model performance on the CAVE dataset. The spectral downsampling process is the same as those in previous experimental settings. The results are shown in Figure 18. We can see that changing $C_{min} = 1$ does not change the trend of the curve compared to Figure 3 and Figure 10, where $C_{min} = 8$. Similarly, we can consistently see that SSIF-RF-GS and SSIF-RF-US outperforms other SSIF variants and the recent baseline LISSF (Zhang et al., 2024).

### A.15 DISCUSSIONS ON SSIF'S GENERALIZATION ACROSS DIFFERENT SPECTRAL BANDS

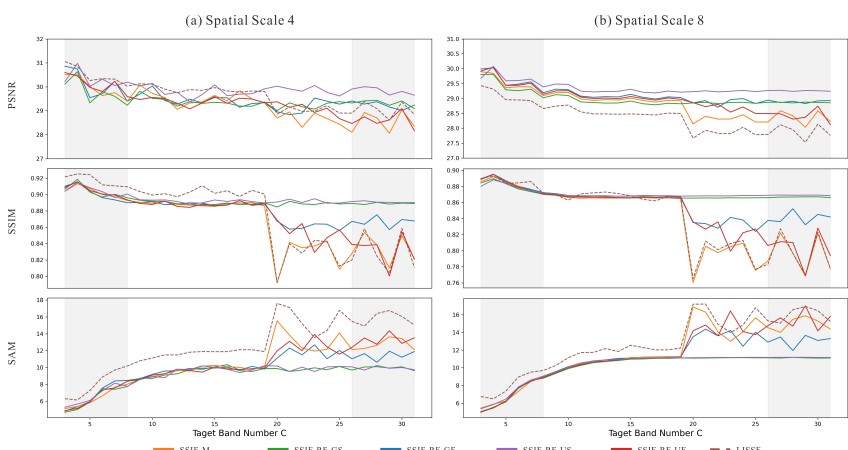

Figure 19: Evaluation results after training different models on truncated training images on the CAVE dataset. Here, we only use the 8-26 bands of the training images of the CAVE dataset as training data for different SR models. Then we test them in the original 1-31 bands of the testing images of the CAVE dataset. The gray area indicates the area of out-of-distribution spectra, i.e., 1-7 and 27-31 bands of the testing images.

In addition, in order to evaluate the generalizability of SSIF across different spectral bands, we do another experiment by truncating the CAVE dataset in the spectral domain. Here, we truncate the 1-7 and 27-31 bands of the training images in the CAVE dataset and train the model using only the 8th to 26th bands of the training images in CAVE. Then we evaluate the trained SR models on the original testing images in the CAVE dataset (containing all 31 bands). Here, we call 8-26 bands in the testing images "in-distribution" data while the 1-7, and 27-31 bands of these images are "out-of-distribution" bands. As shown in Figure 19, SSIF-RF-GS and SSIF-RF-US show consistent performance, i.e., strong model generalizability across spectral space, in both in-distribution and out-of-distribution spectral bands.

Figure 20, and 21 visualize the error maps of LISSF and SSIF on "out-of-distribution" spectral bands. We can see that compared with LISSF, our SSIF demonstrates stronger generalizability across the unseen spectral intervals.

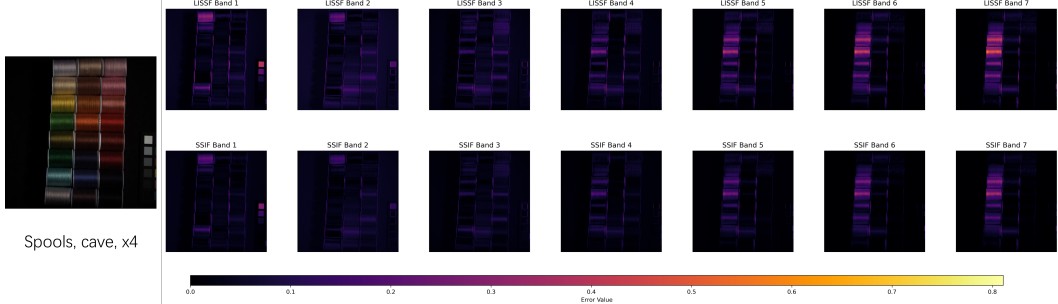

Figure 20: Error maps of LISSF and SSIF on out-of-distribution spectral bands. SSIF and LISSF models are trained on bands 8–26 of the CAVE training images with spatial scales 1–8. Both models are evaluated on the "out-of-distribution" bands, i.e., band 1–7 of CAVE testing images. Here, the spatial SR scale $p = 4$ is within the training spatial SR scales. The error maps are the mean average error.

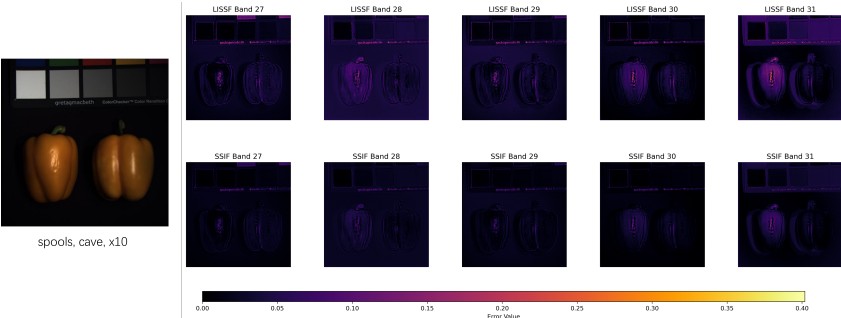

Figure 21: Error maps of LISSF and SSIF on out-of-distribution spectral bands. SSIF and LISSF models are trained on bands 8–26 of the CAVE dataset with scales 1–8. Both models are evaluated on the "out-of-distribution" bands, i.e., bands 27-31, of CAVE testing images. Here, the spatial SR scale $p = 10$ is outside of the training spatial SR scales which is from 1 to 8. The error maps are the mean average errors.

## A.16 MORE VISUALIZATION RESULTS

Figure 22 and 23 present the visual comparison results and corresponding error maps on the CAVE dataset, with a specific focus on the in-distribution spatial scale. The error maps are the mean average errors calculated on the generated RGB bands. The red circles and rectangles in Figures 22 and 23 highlight image regions where our SSIF shows big improvements compared with all baseline models. As evident from these results, SSIF consistently outperforms other baselines, achieving significantly lower error across the entire image.

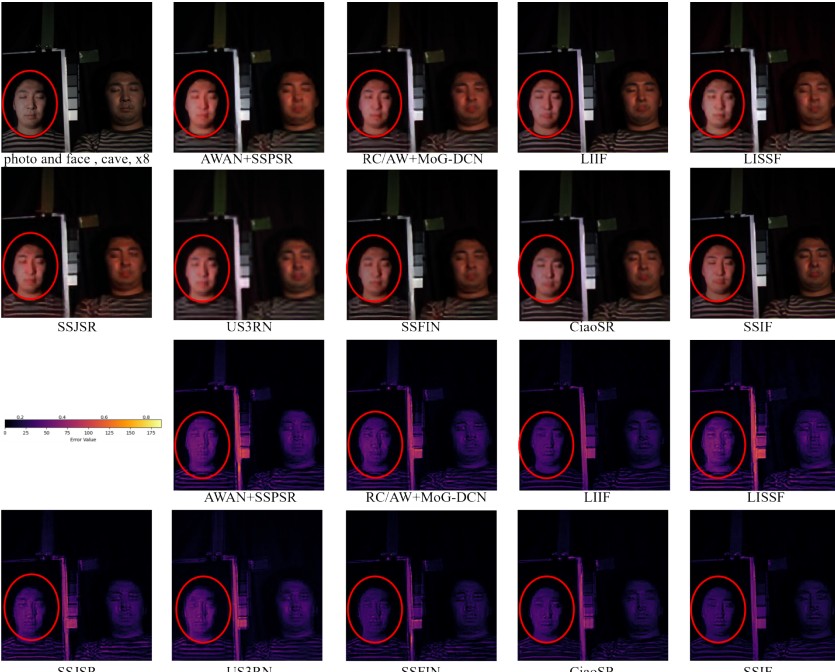

Figure 22: Visual comparison and corresponding error maps on spatial SR on CAVE dataset. Here, the spatial SR scale $p = 8$ which is within the training range of $p$. The error maps are the mean average errors calculated on the RGB bands. The red circle highlights the noticeable improvement of SSIF over other baselines.

Figure 24 provides a comparison of different SR models' SSSR results ($p = 4$) in the spectral dimension, where the error maps are computed across all reconstructed 31 bands. We can see that SSIF demonstrates superior performance compared to other models.

Figures 25 compares the visual results for out-of-distribution evaluation across different spatial scales $p = 10, 14$. Among all baselines, SSIF achieves the best performance in these challenging settings, showing its robustness and generalization capabilities.

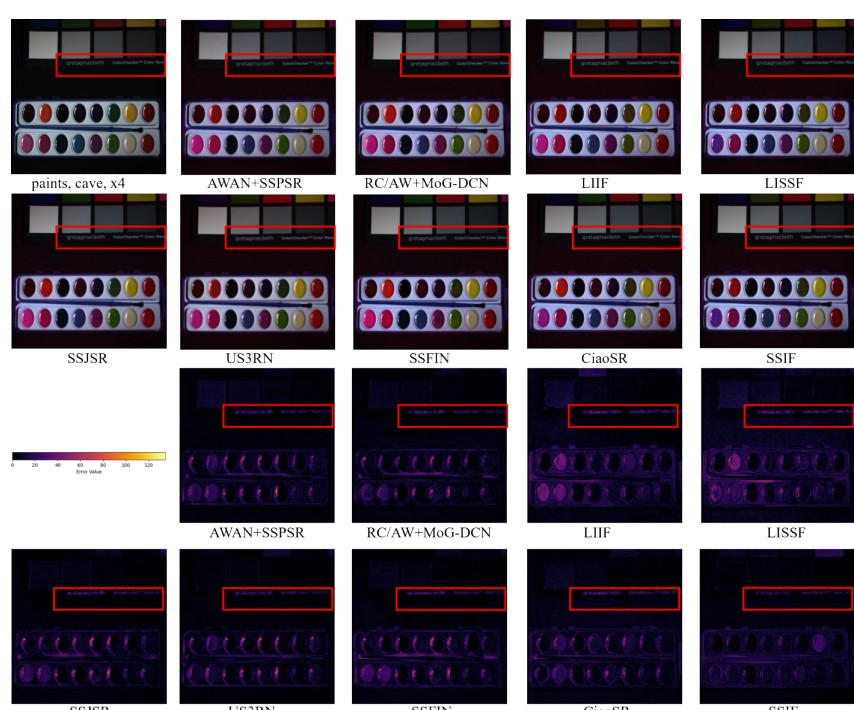

Figure 23: Visual comparison and corresponding error maps on spatial SR on the CAVE dataset. Here, the spatial SR scale $p = 4$ which is within the training range of $p$. The error maps are the mean average errors calculated on the RGB bands. The red rectangle highlights the noticeable improvement of SSIF over other baselines.

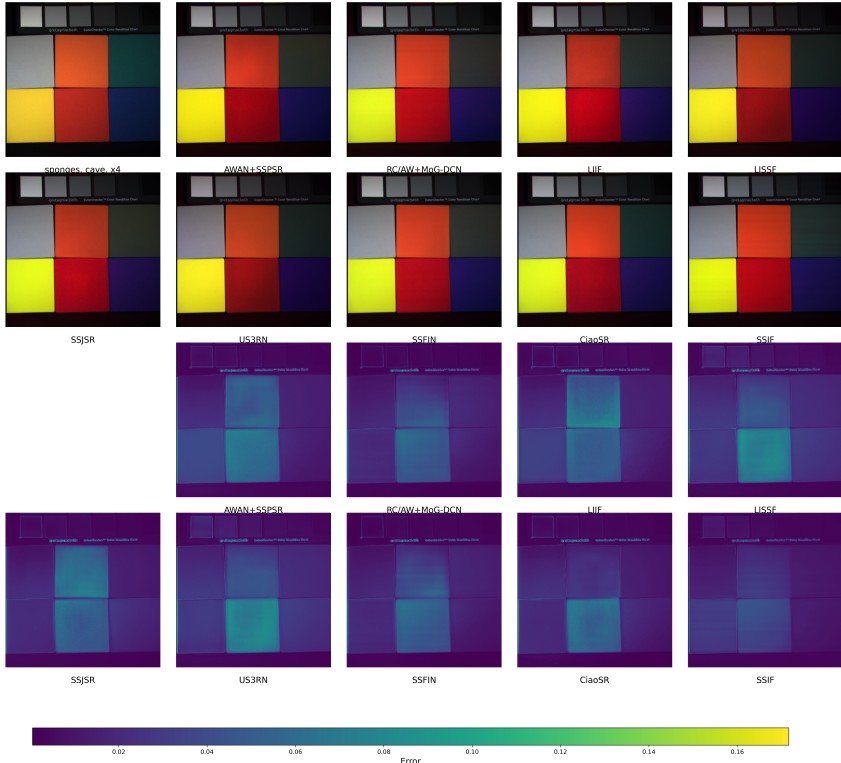

Figure 24: Visual comparison and corresponding error maps of different SR models on the CAVE dataset. Here, the spatial SR scale $p = 4$ which is within the training range of $p$. Instead of showing MAE on the RGB bands in 22 and 23, these error maps are the mean average error calculated on the reconstructed 31 bands.

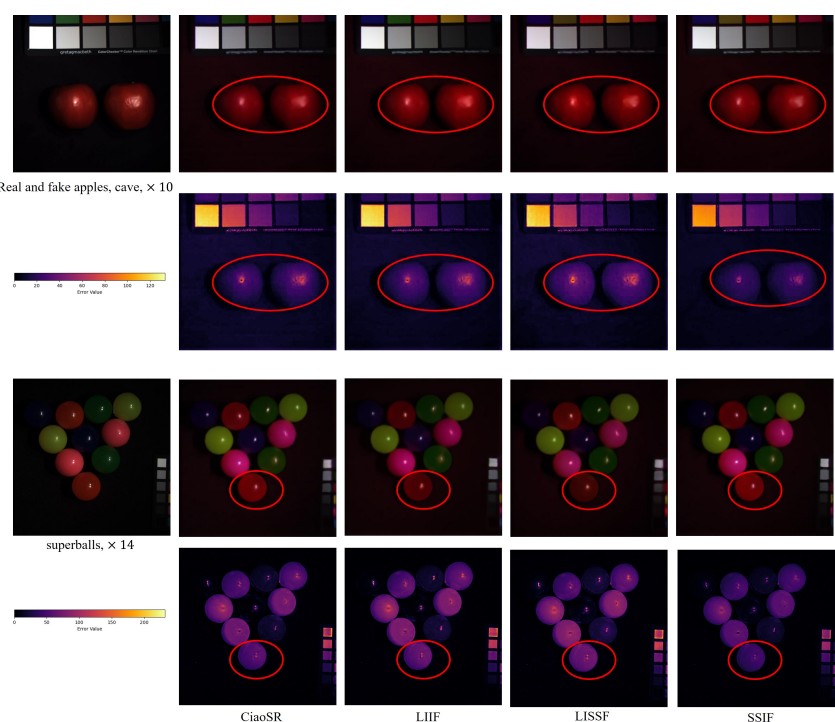

Figure 25: Visual result comparison of different models on out-of-distribution spatial scales, i.e., $p = 10, 14$. All models are trained on spatial scales ranging from 1 to 8. The evaluation is conducted on $p = 10, 14$, which are outside of the training spatial SR scales. The error maps are the mean average errors calculated on the RGB bands. Each column indicates one SR model.

