# OpenReview forum: "SSIF: Physics-Inspired Implicit Representations for Spatial-Spectral Image Super-Resolution"
_ICLR.cc/2025/Conference — Submitted to ICLR 2025_

### Official Review · Reviewer_xER1 · 2024-10-30

**Soundness:** 3
**Presentation:** 2
**Contribution:** 3
**Rating:** 6
**Confidence:** 5

**Summary:**

This paper utilizes an implicit representation framework to address both spatial and spectral degradation in hyperspectral restoration tasks. Various structural variants are proposed based on different sampling strategies, achieving promising results in both in-distribution and out-of-distribution scenarios.

**Strengths:**

The motivation of this study is articulated clearly, beginning with the physical imaging process of hyperspectral sensors to design spectral super-resolution. Experiments confirm the effectiveness of this design. Spatial super-resolution has already been widely adopted in previous work, and this paper combines both approaches to achieve dual-domain super-resolution.

**Weaknesses:**

1、This paper has undergone multiple revisions over an extended period, but it still requires the addition of experiments of 2024. \
2、Implicit representations have become quite advanced in the task of spatial super-resolution, and the approach presented in this paper is relatively straightforward. However, this work lacks a certain level of innovation in the spatial dimension of super-resolution.\
3、This paper still contains typographical errors, such as in section A.5 'SSIF Model Variants,' where 'Both' on line 978 should be corrected to 'both.' Additionally, while the paper presents numerous experiments, some, such as those in section A.14, lack significance. For instance, the careful selection of only two points does not effectively demonstrate the model's superiority.

**Questions:**

1、How are the starting and ending points, $\lambda_{i,s}$ and $\lambda_{i,e}$, determined for each wavelength interval?" \
2、How are the points within the wavelength interval selected? Does the choice between random and non-random selection significantly affect the results?\
3、To investigate how the selection of points affects the results, would it be possible to adopt a coarse-to-fine approach, similar to NeRF, to re-sample the points?\
4、Why is the variance in a Gaussian distribution expressed in this form $\sigma_i = \frac{\lambda_{i,e} - \lambda_{i,s}}{6}$?

---

### Official Review · Reviewer_iYg4 · 2024-11-03

**Soundness:** 2
**Presentation:** 2
**Contribution:** 3
**Rating:** 5
**Confidence:** 5

**Summary:**

This paper proposes the Spatial-Spectral Implicit Function (SSIF), a physics-inspired neural implicit model that represents an image as a continuous function of both pixel coordinates in the spatial domain and wavelengths in the spectral domain. The authors validate SSIF on two challenging benchmarks, demonstrating its superior performance.

**Strengths:**

1. This article presents a relatively novel approach to HSI spatial-spectral super-resolution, attempting to achieve it from the perspective of continuous physical space.
2. SSIF demonstrates good efficiency under low data conditions and converges more quickly during the training process.

**Weaknesses:**

1. The description of the methods section in this paper is insufficient.
2. This article's main innovation lies in combining the physical principles of spectral imaging with the spectral response functions of sensors to achieve HSI spatial-spectral super-resolution. Unfortunately, the authors do not provide sufficient explanation and analysis in the paper.
3. The experimental section lacks the latest relevant methods.

**Questions:**

1. The physical principles of spectral imaging in SSIF’s model design lack deep analysis of their physical property. It is unknown whether the learned neural network reflects physical properties.
2. The author should provide some relevant implicit function-based methods for comparison.
3. The paper seems like a combination of INR, SwinIR, and ciaoSR in HSI spatial-spectral tasks. Could you elaborate on any challenges faced when designing SSIF? This clarification would help in understanding the complexity and novelty of your methodology. Actually, as mentioned in the weaknesses, the explanation of certain blocks in this paper lacks soundness.

---

### Official Review · Reviewer_tQQy · 2024-11-04

**Soundness:** 3
**Presentation:** 3
**Contribution:** 3
**Rating:** 5
**Confidence:** 3

**Summary:**

This work proposed Spatial-Spectral Implicit Function (SSIF), which generalizes neural implicit representations to the spectral domain as a physics-inspired architecture by incorporating sensors’ physical principles of spectral imaging.This method is quite appealing, but there are still many issues that need to be resolved.

**Strengths:**

1.The figures used to analyze the algorithm are very striking, as shown in Figure 1.
2.Diverse methods were used to analyze the performance of each band in the images.

**Weaknesses:**

1.Table 1 lacks a consistent number of decimal places; for example, 9.3, 13.2, and 27.3 have one decimal place, while the remaining values have two.
2. The latest experiment only includes one comparison algorithm for 2023; a comparison algorithm for 2024 should be added.
3.The authors conducted experiments on arbitrary channels with 𝐶>31 in CAVE, but the total number of channels in CAVE is only 31. My suggestion is to see if it is possible to recover the channel count from 𝐶<31 back to 31, and then perform comparative experiments.
4.Figure 19 could benefit from the addition of an error plot to better illustrate the comparison results

**Questions:**

1.Is the comparison algorithm from the most recent year?
2.Channel analysis should be conducted under the guidance of the ground truth (GT); in other words, I would like to see the channels reshaped from C<31 to 31

---

### Official Review · Reviewer_nBSh · 2024-11-04

**Soundness:** 3
**Presentation:** 3
**Contribution:** 3
**Rating:** 5
**Confidence:** 5

**Summary:**

This paper introduces a neural implicit model that expresses an image as a function of continuous pixel coordinates in the spatial domain and continuous wavelengths in the spectral domain, achieving simultaneous super-resolution in both dimensions—an approach that has not been explored before. The study demonstrates substantial innovation, is well-supported by theory, and includes extensive comparative and ablation experiments. In summary, this paper presents impressive innovations and demonstrates substantial effort. However, *Issue 1* raise some doubts about the reliability of the experimental results. Moreover, since the performance in spectral super-resolution is more appealing, the related experiments need improvement (such as Issue 3). If the concerns I raised are addressed, I would consider raising the score.

**Strengths:**

This paper introduces a neural implicit model that expresses an image as a function of continuous pixel coordinates in the spatial domain and continuous wavelengths in the spectral domain, achieving simultaneous super-resolution in both dimensions—an approach that has not been explored before. The study demonstrates substantial innovation, is well-supported by theory, and includes extensive comparative and ablation experiments.

**Weaknesses:**

1.	Using the CAVE dataset as an example, CAVE contains 31 spectral bands, meaning it lacks ground truth (GT) data for cases where C > 31. Therefore, how was the PSNR for C > 31 in Figure 3 calculated? If the GT is obtained through interpolation, wouldn't the model's performance be limited to only approximating the performance of the interpolation?
2.	The paper explains how \(C\) is selected, but how is the channel dimension \(c\) of the HR-MSI (with a shape of \(H \times W \times c\)) determined? In Figure 3, the spectral bands from 1 to 7 are considered out-of-distribution. Is this because \(c\) is chosen from the range of 8 to 31? Was the downsampling performed by simply truncating consecutive spectral bands, or was it uniformly sampled across the range? The lack of clarity in this explanation makes it unclear why \(C_{\min} = 8\) was chosen instead of 1.
3.	For the out-of-distribution spectra, I would prefer to see a visual representation of the results, but this was not included in the paper.
4.	Since \(\Lambda\) can be specified, is it possible to achieve spectral compression, such as converting an HSI image into an RGB image?

**Questions:**

1.	Using the CAVE dataset as an example, CAVE contains 31 spectral bands, meaning it lacks ground truth (GT) data for cases where C > 31. Therefore, how was the PSNR for C > 31 in Figure 3 calculated? If the GT is obtained through interpolation, wouldn't the model's performance be limited to only approximating the performance of the interpolation?
2.	The paper explains how \(C\) is selected, but how is the channel dimension \(c\) of the HR-MSI (with a shape of \(H \times W \times c\)) determined? In Figure 3, the spectral bands from 1 to 7 are considered out-of-distribution. Is this because \(c\) is chosen from the range of 8 to 31? Was the downsampling performed by simply truncating consecutive spectral bands, or was it uniformly sampled across the range? The lack of clarity in this explanation makes it unclear why \(C_{\min} = 8\) was chosen instead of 1.
3.	For the out-of-distribution spectra, I would prefer to see a visual representation of the results, but this was not included in the paper.
4.	Since \(\Lambda\) can be specified, is it possible to achieve spectral compression, such as converting an HSI image into an RGB image?

---

### Comment · Reviewer_iYg4 · 2024-11-26
**Rebuttal**

Thank you for ICLR's message regarding the ongoing discussion. I'd like to have a quality discussion, but the author seems to have given up on this. I regret this.

---

> ### Author Response · Authors · 2024-11-26
> **Response to Reviewer iYg4**
>
> We would like to express our gratitude to Reviewer iYg4 for the reminder. We have been diligently working to incorporate the additional experiments as suggested. Preparing the new experimental results required some time, but we have now updated our submission in response to your valuable suggestions. We greatly appreciate your feedback.

---

### Meta-Review · Area_Chair_675B · 2024-12-17

**Metareview:**

The paper proposes Spatial-Spectral Implicit Function (SSIF), a neural implicit model that represents an image as a function of both continuous pixel coordinates in the spatial domain and continuous wavelengths in the spectral domain. The motivation of this study is articulated clearly. However, I agree with the reviews’ comments that the description of the methods section in this paper is insufficient. It is unclear what advantages the proposed method offers compared to other hyperspectral image super-resolution methods based on implicit functions.
Based on the reviewer's average score, I believe that this paper is not yet ready for publication at ICLR.

**Additional Comments On Reviewer Discussion:**

Reviewer xER1 has raise the score due to the authors' response, and the additional experiments.
The author has also responded to the questions raised by other reviewers, yet the overall quality of the paper still appears to be insufficient.

---

### Decision · Program_Chairs · 2025-01-22

Reject